



# Shape dependence of snow crystal fall speed

Sandra Vázquez-Martín[1], Thomas Kuhn[1], and Salomon Eliasson[2]

[1]Luleå University of Technology (LTU). Department of Computer Science, Electrical and Space Engineering. Division of Space Technology, 98 128, Kiruna, Sweden.
[2]Swedish Meteorological and Hydrological Institute (SMHI), 601 76, Norrköping, Sweden.

**Correspondence:** Thomas Kuhn (thomas.kuhn@ltu.se)

**Abstract.**

Improved snowfall predictions require accurate knowledge of the properties of ice crystals and snow particles, such as their size, cross-sectional area, shape, and fall speed. In particular, the shape is an important parameter as it strongly influences the scattering properties of these ice particles, and thus their response to remote sensing techniques such as radar measurements.

The fall speed of ice particles is a critical parameter for the representation of ice clouds and snow in atmospheric numerical models, as it determines the rate of removal of ice from the modelled clouds. They are also required for snowfall predictions alongside other properties such as ice particle size, cross-sectional area, and shape. For example, shape is important as it strongly influences the scattering properties of these ice particles, and thus their response to remote sensing techniques.

This work analyses fall speed as a function of shape and other properties using ground-based in-situ measurements. The
measurements for this study were done in Kiruna, Sweden during the snowfall seasons of 2014 to 2019, using the ground-based in-situ instrument Dual Ice Crystal Imager (D-ICI). The resulting data consist of high-resolution images of falling hydrometeors from two viewing geometries that are used to determine size (maximum dimension), cross-sectional area, area ratio, orientation, and the fall speed of individual particles. The selected dataset covers sizes from about 0.06 to 3.2 mm and fall speeds from 0.06 to 1.6 m s$^{-1}$.

The particles are shape-classified into 15 different shape groups depending on their shape and morphology. For these 15 shape groups relationships are studied, firstly, between size and cross-sectional area, then between fall speed and size or cross-sectional area. The data show in general low correlations to fitted fall-speed relationships due to large spread observed in fall speed. After binning the data according to size or cross-sectional area, correlations improve and we can report reliable parameterizations of fall speed vs. size or cross-sectional area for part of the shapes. The effects of orientation and area ratio
on the fall speed are also studied, and measurements show that vertically orientated particles fall faster on average. However, most particles for which orientation can be defined fall horizontally.

**Keywords:** Natural snow crystals; hydrometeors; microphysical properties; fall speed; orientation; ground-based in-situ measurements.





## 1 Introduction

The accurate knowledge of the microphysical properties of atmospheric snow particles (ice crystals and snowflakes) is essential to achieve more realistic parameterizations in atmospheric models (e.g., Stoelinga et al., 2003; Tao et al., 2003). Also, the accuracy of many different remote sensing applications, such as scattering properties, cloud and precipitation retrievals from satellite passive and active microwave measurements (e.g., Bauer et al., 1999; Olson et al., 2001), and snowfall estimates based

on ground- and space-based radar (Cooper et al., 2017), is highly dependent on the assumptions made on the microphysical properties of snow particles. Some of these critical properties are particle size, particle size distribution, cross-sectional area, area ratio, shape, orientation, and fall speed.

Several studies have classified snow crystal shape according to different classification schemes (Nakaya, 1954; Magono and Lee, 1966; Kikuchi et al., 2013; Vázquez-Martín et al., 2020). Particle shape is essential, not only for investigating growth

processes but also because it affects remote sensing measurements, for instance, radar measurements (e.g., Sun et al., 2011; Matrosov et al., 2012; Marchand et al., 2013) or passive measurements of microwave brightness temperatures (Kneifel et al., 2010). Furthermore, it is of significant importance for optical remote sensing retrievals of cloud properties (see, e.g., Yang et al., 2008; Baum et al., 2011; Xie et al., 2011; Loeb et al., 2018) and snow albedo (e.g., Jin et al., 2008). The physical properties of snow particles, including shape, govern their fall speed. For a given volume and density, non-spherical particles

fall slower than spheres (Haider, 1989). Therefore, also the particle shape is an important parameter to ensure accurate cloud parameterizations in climate and forecast models (e.g., Stoelinga et al., 2003; Tao et al., 2003), and for the understanding of precipitation in cold climates.

Together with particle size and shape, the snow particle orientation also plays a significant role. It is highly dependent on the local aerodynamic conditions (Pruppacher and Klett, 1997), and significant uncertainties regarding particle orientation

remain, especially for rimed particles and aggregates (Xie et al., 2012; Jiang et al., 2019). The particle orientation determines its horizontal cross-sectional area and influences its drag, and, therefore, its fall speed. Particle orientation also affects the bulk scattering properties of clouds (Yang et al., 2008, 2011). For instance, for microwave radiation, particle orientation significantly affects the radar reflectivity (e.g., Sun et al., 2011; Gergely and Garrett, 2016), and due to its sizeable impact on absorption (e.g., Foster et al., 2000), strongly modulates the microwave brightness temperature (Xie and Miao, 2011; Xie et al., 2015).

The fall speed of snow crystals plays a significant role in modelling microphysical precipitation processes (Schefold et al., 2002) and for climate since it determines the lifetime of cirrus clouds and influences the vertical transport of water vapour in the upper troposphere, modulating the top of atmosphere radiation budget (Westbrook and Sephton, 2017). Additionally, fall speed determines the snowfall rate, i.e., the rate of particle removal from clouds. The precipitation rate is proportional to the fall speed of the particles, implying quantitative forecasts of this variable require accurate snowflake fall speeds (Westbrook

and Sephton, 2017). Therefore, it is essential to know particle size, shape, and fall speed simultaneously.

Earlier studies have used different methods to investigate and parameterize the dependence of fall speed on snow particle size. Most parameterizations can be given as a power law with general form $v = a_D \cdot D^{b_D}$, where $v$ is the fall speed, $D$ is the particle size, and $a_D$, $b_D$ are constant coefficients. This power-law relationship is often adopted because it facilitates analytical





solutions in models, for instance for calculations of Doppler velocity, and appears in many studies (e.g., Locatelli and Hobbs,
1974; Heymsfield and Kajikawa, 1987; Mitchell, 1996; Barthazy and Schefold, 2006; Yuter et al., 2006; Brandes et al., 2008;
Heymsfield and Westbrook, 2010; Zawadzki et al., 2010; Lee et al., 2015). The dependence of fall speed on particle cross-
sectional area is also readily represented as a power law, $v = a_A \cdot A^{b_A}$, where $v$ is the fall speed, $A$ is the cross-sectional
area, and $a_A$, $b_A$ are constant coefficients (e.g., Kuhn and Gultepe, 2016; Kuhn and Vázquez-Martín, 2020). In a few studies,
different functions are used to describe relationships. For example, Barthazy and Schefold (2006) showed that an exponential
function that asymptotically approaches a constant speed at larger sizes could also be used to describe the size dependence of
fall speed, in particular for particles larger than about 3 mm.

This study analyses the fall speed relationships of snow particles as a function of particle size and cross-sectional area
based on a dataset of falling natural snow particles that have been collected in Kiruna in northern Sweden with the ground-
based instrument Dual Ice Crystal Imager (D-ICI) presented in Kuhn and Vázquez-Martín (2020). Section 2 describes the
measurements and methods used to classify the snow particle shape and determine their size, cross-sectional area, and fall
speed. Section 3 investigates the influence of the particle shape on relationships between fall speed and particle size or cross-
sectional area. Furthermore, we examine the dependence of fall speed on area ratio and particle orientation. These results are
then compared to previous studies. Finally, this study is summarized and concluded in Sect. 4.

## 2 Methods

### 2.1 Measurements and instrument

Our measurements are carried out in Kiruna, Sweden (67.8° N, 20.4° E) using D-ICI, the ground-based in-situ instrument
described in Kuhn and Vázquez-Martín (2020). D-ICI captures and records dual images of falling snow crystals and other
hydrometeors. Detected particles are imaged simultaneously from two different viewing directions. One is horizontal, recording
a side view, and one is close to vertical, recording a top view. From the top-view image, we can determine particle size, cross-
sectional area, and area ratio. From the side-view image, since it is exposed twice, we can determine fall speed (see Sect. 2.2).

These images have a high optical resolution of about 10 µm, and one pixel corresponds to 3.7 µm. This resolution allows
for the identification of snow particles even smaller than 0.1 mm. The additional information dual images provide, improves
the shape classification carried out by looking at both top- and side-view images. The particles are classified according to their
shape and sorted into 15 shape groups, as described in Vázquez-Martín et al. (2020).

More than 10,000 particles have been recorded during multiple snowfall seasons, the winters of 2014/2015 to 2018/2019.
Each winter season at the Kiruna site lasts approximately from the beginning of November to the middle of May. Part of these
data from 2014 to 2018 was selected to carry out this work. During image processing (see Sect. 2.2), we only consider particles
that are entirely in the field of view and that are not significantly tumbling (for a detailed description, see Kuhn and Vázquez-
Martín, 2020). Furthermore, the ambient wind speed is taken into account. As mentioned in Kuhn and Vázquez-Martín (2020),
higher wind speeds may alter fall speed measurements. Therefore data taken at averaged wind speeds higher than 3 m s$^{-1}$ are
excluded. The SMHI weather data (SMHI, 2020), based on instruments at the Kiruna airport, are recorded 6 km away from our





measurement site and provide the wind speed data used in this study. The site in Kiruna does not often experience high wind speeds; hence the dataset is only reduced by 23% to 2,461 particles from 48 days.

## 2.2 Snow properties

Figure 1 shows two different snow particles from the side *(right)* and the top view *(left)*. The images from the top view are used to determine particle size, cross-sectional area, and area ratio by the automated process presented in Kuhn and Vázquez-Martín (2020). For this, first, the background features are removed, then the in-focus particles are detected, and their boundaries traced. Consequently, the particle properties, such as particle size, cross-sectional area, and area ratio can be determined. As we have described in Vázquez-Martín et al. (2020), the maximum dimension, $D_{max}$, defined as the smallest diameter that

completely encircles the particle boundary in the top-view image, is used to describe the particle size. The cross-sectional area, $A$, is defined as the area in the top-view image enclosed by the particle boundary based on pixel count. Once, particle size and cross-sectional are determined, the area ratio $A_r$ can be also calculated from these quantities:

$$A_r = \frac{A}{\frac{\pi}{4} \cdot D_{max}^2}. \tag{1}$$

These quantities are particularly relevant when provided from this vertical viewing geometry corresponding to the falling

motion, rather than a horizontal geometry, which is standard in many instruments.

The side-view images are exposed twice to enable fall speed measurements so that both particle exposures are displayed in the same image (Fig. 1, *right*). These particle exposures correspond to the first and second position, respectively, of the particle when falling. In our data, the two-particle exposures in the side-view images might be partly overlapping due to a combination of fall speed and size of the particle. Figure 1a shows an example of non-overlapping particles, whereas, in Fig. 1b, the particles

are partly overlapping, which poses a limitation for an automated fall speed determination. The following describes the manual procedure to, nonetheless, include such particles in the analysis.

At least two points of the particle need to be selected, for instance, the left and right edges of the particle ($P_1$ and $P_3$ in Fig. 1, *right*). The same points are found by eye on the second exposure ($P_2$ and $P_4$ in Fig. 1, *right*). The falling distance is then the average of the euclidean distances between $P_1$ and $P_2$, and between $P_3$ and $P_4$, and the fall speed is this falling

distance divided by the time between exposures. By selecting at least two points on each particle to determine fall speed, one can notice differences of the fall speed across the particle. If there is no difference, then the particle is falling straight. If there is a difference, then the particle is tumbling, i.e., has a rotating motion in addition to the straight falling motion. Tumbling is most noticeable if the rotation is around an axis perpendicular to the imaging plane.

When rotating around an axis parallel to the imaging plane, it may be challenging to select the same points on the second

exposure. Particle images where it is difficult to identify the same points on both exposures, or when significant tumbling is apparent, are excluded. The tumbling limit is when the speed of the points differ by more than $\pm 10\%$ from the mean speed. However, tumbling is not observed frequently in our dataset. Figure 2 shows different side-view images of particles included and excluded from the analysis, respectively. In Fig. 2a–d, the particles are not, or are only slightly, tumbling, and therefore



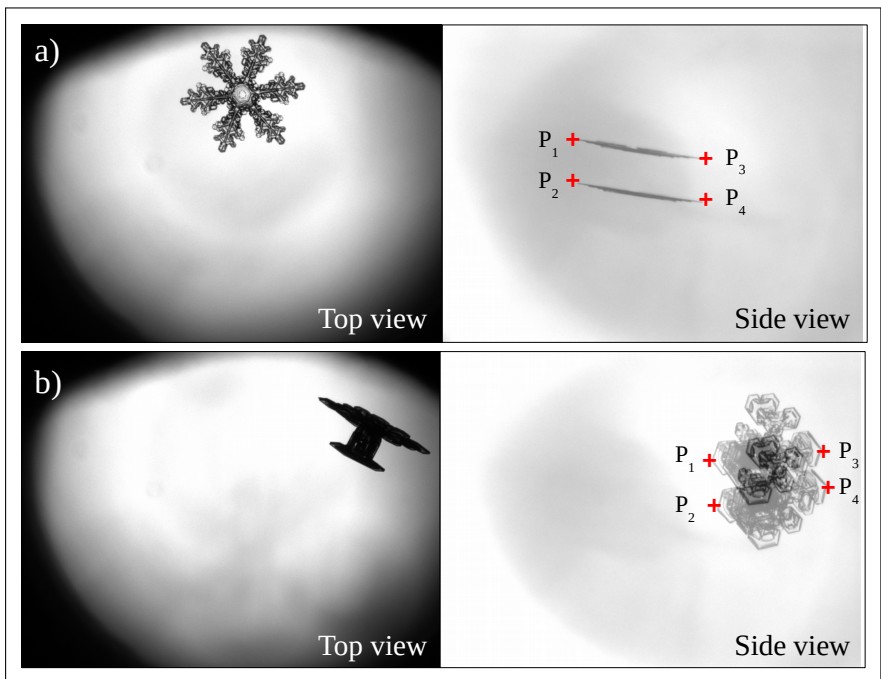

**Figure 1.** Two particle examples **(a–b)**. *Left:* Top-view images. *Right:* Side-view images, which are exposed twice to enable fall speed measurements. Both viewing geometries are used to classify the particle shape. In both examples, two pairs of points ($P_1$, $P_3$ and $P_2$, $P_4$) were selected to determine the fall speed.

included in the analysis. In Fig. 2e–h, the particles are tumbling significantly. Consequently, these particles were discarded and
not included in the analysis.

## 3   Results and discussions

### 3.1   Cross-sectional area

For this study, we use a large subset of the data from Vázquez-Martín et al. (2020). Although we excluded measurements with higher wind speeds than $3\,\mathrm{m\,s^{-1}}$, the cross-sectional areas as a function of particle size are nonetheless very similar here
to results presented in Vázquez-Martín et al. (2020). However, for completeness, we briefly analyse and discuss the cross-sectional area versus particle size, $A$ vs. $D_{\max}$, for all the shape groups in this section. Table 1 shows these results, along with the meta-data on the particle groups, including their full names. For simplicity, we will use shorter names from here on (see, e.g., in Table 2). As seen in Table 1, generally, particle size and cross-sectional area are very well correlated ($R^2 > 0.7$) if expressed by the power law

$$A(D_{\max}) = a \cdot \left( \frac{D_{\max}}{1\,\mathrm{mm}} \right)^b,$$     (2)





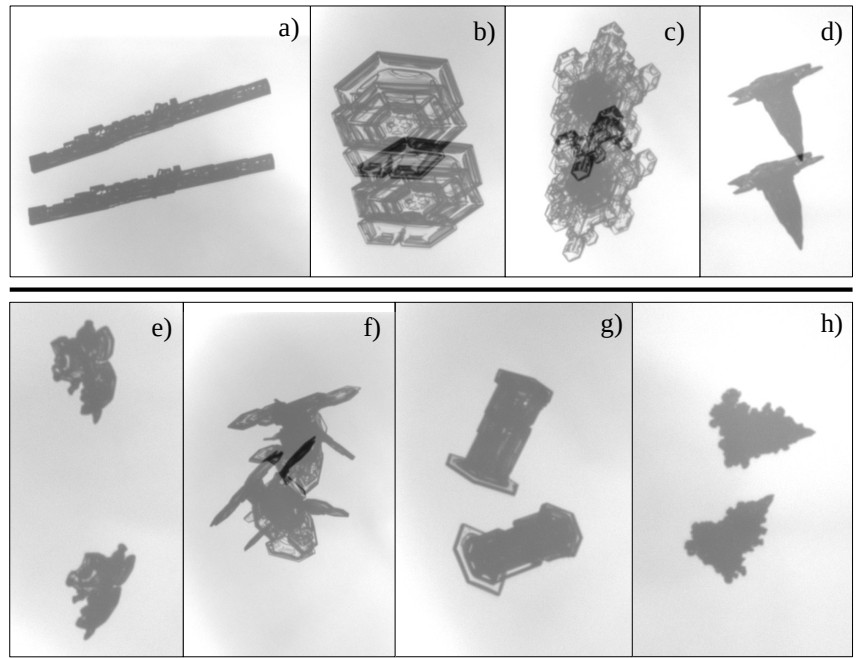

**Figure 2.** Panels **(a–h)** show eight examples of different particles with side-view images. The panels in the top row **(a–d)**, show particles that were included in the analysis. Panels in the bottom row **(e–h)**, show particles that were excluded since the two-particle exposures reveal significant tumbling.

where the parameter $a$ corresponds to the cross-sectional area at $D_{\max} = 1$ mm and $b$ is the exponent in the power law.

Figure 3 shows these fitted $A$ vs. $D_{\max}$ relationships. We note that shape groups *(1) Needles*, *(2) Crossed needles*, and *(3) Thick columns* are the groups with the lowest values of parameter $b$ that are close to 1. For these groups, this is understandable from their morphology. An increase in $A$ primarily follows an increase in $D_{\max}$ (needle length), rather than in both $D_{\max}$ and the diameter (needle width). The low values of $b$ also explain why the area ratio, $A_{\mathrm{r}}$, decreases most rapidly with increasing $D_{\max}$ for these shape groups, which can be seen if one expresses $A_{\mathrm{r}}$ as a power law in $D_{\max}$ (inserting Eq. (2) in (1)),

$$A_{\mathrm{r}} = \frac{4}{\pi} \cdot \frac{a}{1\,\mathrm{mm}^2} \cdot \left(\frac{D_{\max}}{1\,\mathrm{mm}}\right)^{b-2}, \tag{3}$$

as the exponent in this power law is $b-2$. It is also evident in Fig. 4, which shows these power laws for all shape groups determined from Eq. (3) and the coefficients given in Table 1.

For most other shape groups, the coefficient $b$ varies between 1.4 and 1.8. Only for the groups *(12) Graupel* and *(15) Spherical*, it is larger with $b = 2.0$, which is the expected theoretical value for spherical shapes. Thus, apart from *(15) Spherical*, *(12) Graupel* is the only shape, for which shape does not change with size, i.e., $A_{\mathrm{r}}$ does not decrease with increasing $D_{\max}$ but remains constant.



**Table 1.** Cross-sectional area versus particle size ($A$ vs. $D_{\max}$) relationship fitted to a power law given by Eq. (2) for each shape group and for all data, i.e., for all the particles regardless of shape. The number of particles $N$, $D_{\max}$, and $A$ ranges (min, max), the parameters $a$ and $b$ with their respective uncertainties, and the correlation coefficients $R^2$ are shown for each shape group and all data.

| Shape groups (1–15) | $N$ | $D_{\max}$ [mm] | $A$ [mm$^2$] | $A$ vs. $D_{\max}$ | | |
|---|---|---|---|---|---|---|
| | | | | $a$ [mm$^2$] | $b$ | $R^2$ |
| (1) Needles and thin or long columns | 317 | 0.27–3.0 | 0.03–0.7 | $0.15 \pm 0.01$ | $1.06 \pm 0.03$ | 0.79 |
| (2) Crossed needles and crossed columns | 66 | 0.57–2.6 | 0.10–0.7 | $0.18 \pm 0.04$ | $1.01 \pm 0.08$ | 0.70 |
| (3) Thick columns and bullets | 103 | 0.16–0.9 | 0.02–0.2 | $0.17 \pm 0.04$ | $1.24 \pm 0.05$ | 0.88 |
| (4) Capped columns and capped bullets | 189 | 0.28–2.1 | 0.02–1.3 | $0.32 \pm 0.03$ | $1.60 \pm 0.06$ | 0.79 |
| (5) Plates | 197 | 0.21–1.7 | 0.03–1.3 | $0.45 \pm 0.03$ | $1.71 \pm 0.03$ | 0.93 |
| (6) Stellar crystals | 43 | 0.54–2.3 | 0.13–1.9 | $0.40 \pm 0.07$ | $1.59 \pm 0.15$ | 0.75 |
| (7) Bullet rosettes | 41 | 0.54–1.5 | 0.12–0.8 | $0.35 \pm 0.04$ | $1.62 \pm 0.12$ | 0.83 |
| (8) Branches | 438 | 0.27–2.9 | 0.03–3.4 | $0.32 \pm 0.01$ | $1.74 \pm 0.03$ | 0.86 |
| (9) Side planes | 350 | 0.29–2.7 | 0.04–2.7 | $0.37 \pm 0.01$ | $1.77 \pm 0.04$ | 0.87 |
| (10) Spatial plates | 48 | 0.30–1.3 | 0.06–0.6 | $0.42 \pm 0.05$ | $1.62 \pm 0.10$ | 0.85 |
| (11) Spatial stellar crystals | 185 | 0.36–2.8 | 0.06–2.9 | $0.40 \pm 0.01$ | $1.76 \pm 0.03$ | 0.95 |
| (12) Graupel | 37 | 0.25–1.2 | 0.03–0.8 | $0.51 \pm 0.04$ | $1.99 \pm 0.05$ | 0.98 |
| (13) Ice and melting or sublimating particles | 60 | 0.13–1.2 | 0.01–0.3 | $0.23 \pm 0.09$ | $1.45 \pm 0.07$ | 0.87 |
| (14) Irregulars and aggregates | 346 | 0.21–3.2 | 0.02–2.7 | $0.34 \pm 0.02$ | $1.65 \pm 0.03$ | 0.91 |
| (15) Spherical particles | 41 | 0.06–0.4 | 0.003–0.15 | $0.80 \pm 0.02$ | $2.04 \pm 0.01$ | 1.00 |
| **All data** | 2461 | 0.06–3.2 | 0.003–3.4 | $0.30 \pm 0.01$ | $1.54 \pm 0.01$ | 0.81 |

The fitted relationships for all particles (regardless of shape) also appear in Fig. 3 and 4 for $A$ vs. $D_{\max}$ and $A_{\mathrm{r}}$ vs. $D_{\max}$, respectively. They represent a kind of average; however, they do not seem to be a good approximation for most of the shapes.

### 3.2 Fall speed relationships

This section provides an analysis of how the particle size, cross-sectional area, and shape influence its fall speed. Figure 5 shows the distribution of fall speed data for each shape group and all data (regardless of shape). A vertical bar represents the distribution of fall speed data for each shape. The bounds of each bar correspond to the 16$^{\mathrm{th}}$ and 84$^{\mathrm{th}}$ percentiles of the distribution. The point on the bar shows the location of the median value. These bounds would correspond to $\pm 1\sigma$ (standard deviation) if the distribution were normal. Table 2 contains a list of these percentiles and medians. We note that shape groups *(7) Bullet rosettes* and *(12) Graupel* have the fastest fall speeds with a median value of $v \simeq 0.6$ m s$^{-1}$. Followed by shape groups *(4) Capped columns*, *(9) Side planes*, *(11) Spatial stellar*, *(14) Irregulars*, and *(15) Spherical* with a median fall speed value of $v \simeq 0.5$ m s$^{-1}$. The median of all data is approximately 0.43 m s$^{-1}$, and most shape groups have their median within





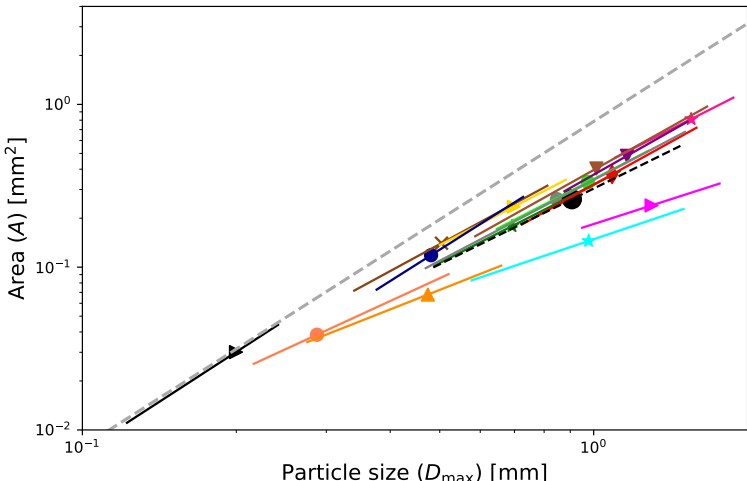

**Figure 3.** Cross-sectional area versus particle size ($A$ vs. $D_{\mathrm{max}}$) relationships are shown in logarithmic scale for all the shape groups (solid lines) and all data (dashed black line). The median $D_{\mathrm{max}}$ of the data is represented as a single point on each line. The length the fit lines are defined by $16^{\mathrm{th}}$ and $84^{\mathrm{th}}$ percentiles of $D_{\mathrm{max}}$. For a legend of the shape groups, see Fig. 5. For comparison, the area of spheres given by $(\pi/4) \cdot D_{\mathrm{max}}^2$ is shown as a grey dashed line.

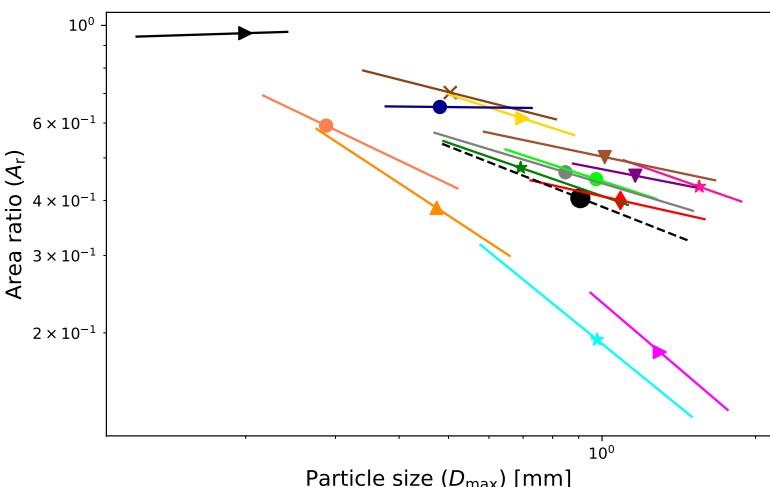

**Figure 4.** Area ratio versus particle size ($A_{\mathrm{r}}$ vs. $D_{\mathrm{max}}$) relationships are shown in logarithmic scale for all the shape groups (solid lines) and all data (dashed black line). The median $D_{\mathrm{max}}$ of the data is represented as a single point on each line. The length the fit lines are defined by $16^{\mathrm{th}}$ and $84^{\mathrm{th}}$ percentiles of $D_{\mathrm{max}}$. For a legend of the shape groups, see Fig. 5.





$\pm 0.08 \, \mathrm{m \, s^{-1}}$ from this value. Shape groups *(1) Needles*, *(2) Crossed needles*, and *(3) Thick columns* have the lowest median values of $0.34 \, \mathrm{m \, s^{-1}}$ or less.

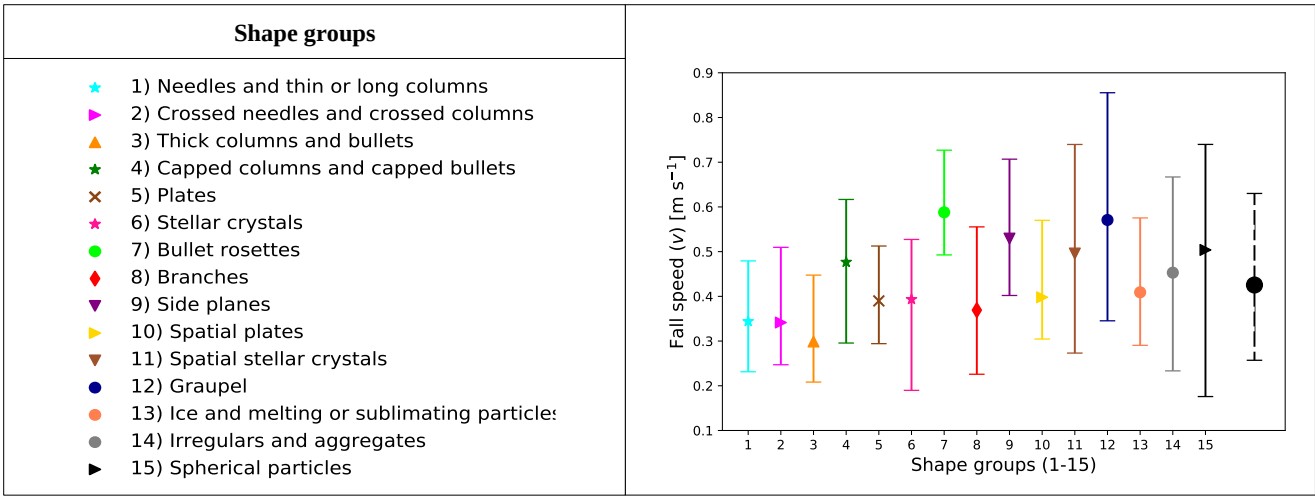

**Figure 5.** The fall speeds $v$ of snow crystals for each shape group are shown in solid lines. The median and the distribution of $v$ are shown. The values of the median are represented as points. The lower and upper ends of the vertical bars indicating the distributions are given by the $16^{\mathrm{th}}$ and $84^{\mathrm{th}}$ percentiles, respectively. For comparison, a black dashed line shows all data (regardless of shape).

### 3.2.1  Fitting to individual data: $M_{\mathrm{a}}$

Fall speed versus particle size ($v$ vs. $D_{\mathrm{max}}$) and fall speed versus cross-sectional area ($v$ vs. $A$) relationships are useful to parameterize fall speed. In order to find the $v$ vs. $D_{\mathrm{max}}$ relationships on the data, one can apply a power-law fit given by

$$v(D_{\mathrm{max}}) = a_D \cdot \left( \frac{D_{\mathrm{max}}}{1 \, \mathrm{mm}} \right)^{b_D} , \tag{4}$$

which yields straight lines on a logarithmic plot, and, from these lines, derive the parameters $a_D$ and $b_D$. The $v$ vs. $A$ relationships result from fitting data to a power law given by

$$v(A) = a_A \cdot \left( \frac{A}{1 \, \mathrm{mm^2}} \right)^{b_A} . \tag{5}$$

The parameters $a_A$ and $b_A$ are determined from linear fits to the data expressed as $\log(v)$ versus $\log(A)$. This method of fitting

to individual data is further referred to as $M_{\mathrm{a}}$. As an example, we look at shape group *(5) Plates*, representing a commonly occurring shape that has clear results. The individual data points of the measured fall speeds appear in Fig. 6. It also shows the 68% prediction band, which describes the region, where to expect 68% of any new measurements. The prediction band can be considered a measure of the spread of the data around the fit, which appears as lines in the same colour as the individual data points.



**Table 2.** Fall speed, $v$, for the shape groups. The fall speed range, the median and $16^{th}$ and $84^{th}$ percentiles are displayed. For comparison, results for all the data, regardless of shape, are also shown.

| Shape groups (1–15) | $v\,[\mathrm{m\,s^{-1}}]$ | | | |
| --- | --- | --- | --- | --- |
| | Range | Median | $16^{th}$ | $84^{th}$ |
| *(1) Needles* | 0.10–0.8 | 0.34 | 0.23 | 0.48 |
| *(2) Crossed needles* | 0.17–0.9 | 0.34 | 0.25 | 0.51 |
| *(3) Thick columns* | 0.14–0.8 | 0.30 | 0.21 | 0.45 |
| *(4) Capped columns* | 0.11–1.0 | 0.48 | 0.30 | 0.62 |
| *(5) Plates* | 0.11–0.9 | 0.39 | 0.29 | 0.51 |
| *(6) Stellar* | 0.13–0.8 | 0.39 | 0.19 | 0.53 |
| *(7) Bullet rosettes* | 0.15–0.8 | 0.59 | 0.49 | 0.73 |
| *(8) Branches* | 0.06–1.2 | 0.37 | 0.23 | 0.56 |
| *(9) Side planes* | 0.19–0.9 | 0.53 | 0.40 | 0.71 |
| *(10) Spatial plates* | 0.16–1.0 | 0.40 | 0.30 | 0.57 |
| *(11) Spatial stellar* | 0.13–1.1 | 0.50 | 0.27 | 0.74 |
| *(12) Graupel* | 0.26–1.0 | 0.57 | 0.35 | 0.86 |
| *(13) Ice particles* | 0.13–1.0 | 0.41 | 0.29 | 0.58 |
| *(14) Irregulars* | 0.06–1.5 | 0.45 | 0.23 | 0.67 |
| *(15) Spherical* | 0.09–1.6 | 0.50 | 0.18 | 0.74 |
| **All data** | 0.06–1.6 | 0.43 | 0.26 | 0.63 |

The large spread in fall speed apparent from Fig. 6, results in a low correlation to the fit functions. The results for this shape group represent the general features found in all shape groups, i.e., a large spread in fall speed data and relatively low $R^2$ for $M_a$. The $v$ vs. $D_{max}$ and $v$ vs. $A$ relationships for all the shape groups are shown in Figures A1-A2 (see Appendix A). Tables 3 and 4, show these results for $v$ vs. $D_{max}$ and for $v$ vs. $A$, respectively. The only exceptions from the generally low correlations are shape groups *(11) Irregulars*, *(12) Graupel*, and *(15) Spherical* with $R^2 > 0.5$. For all other shape groups, the correlation

coefficients for $M_a$ are $R^2 \lesssim 0.2$ for both $v$ vs. $D_{max}$ and $v$ vs. $A$. Judging by these low $R^2$ values, it is uncertain if the fit functions are representative of the measured data.

### 3.2.2 Fitting to binned data: $M_b$

The spread of fall speed data may be considered random noise, and binning the data should remove this noise to some extent. Therefore, to improve the correlation, the data are first binned into ten particle size or cross-sectional area bins before fitting to

Eq. (4) and (5), respectively, where each bin contains as close to the same number of particles as possible. Therefore, the bin widths are variable and specific to each shape group, and thereby avoid the problem of individual bins having a disproportional





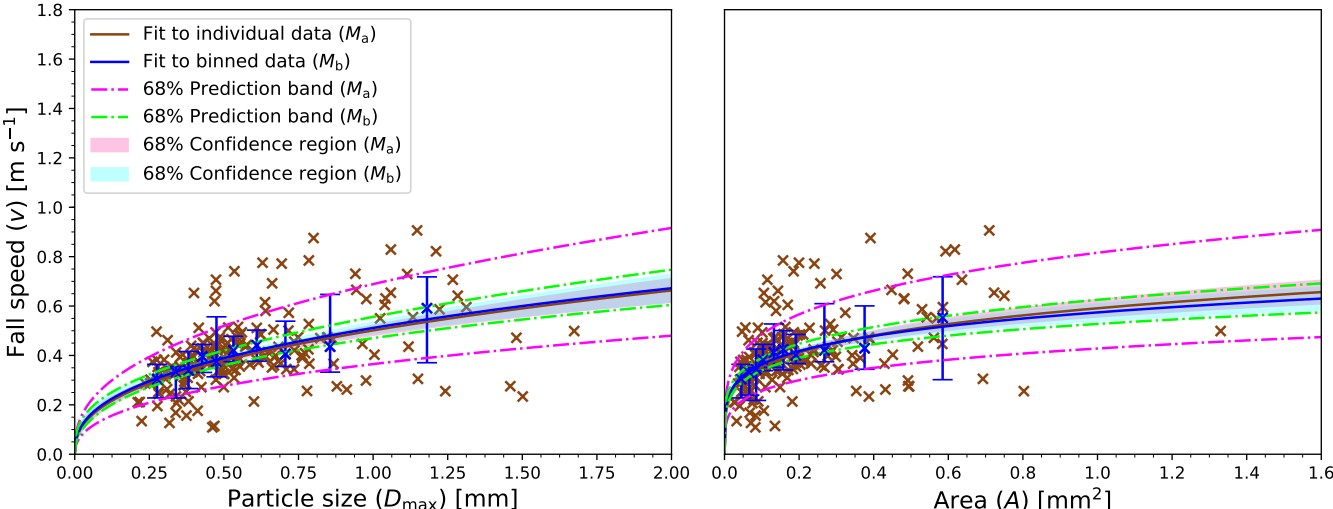

**Figure 6.** Fall speed versus particle size ($v$ vs. $D_{\max}$) and fall speed versus cross-sectional area ($v$ vs. $A$) relationships for shape group *(5) Plates*. Individual data (brown symbols) and binned data (blue symbols with error bars) are displayed. Median values in the respective bins represent the binned data. The total length of the error bars represents the spread in fall speed data, which is given by the difference between the 16$^{\text{th}}$ and 84$^{\text{th}}$ percentiles. Fits that apply to individual data ($M_{\text{a}}$) and to binned data ($M_{\text{b}}$) are shown for comparison. The 68% prediction band and the 68% confidence region for both fits ($M_{\text{a}}$, $M_{\text{b}}$) are also shown. *Left:* $v$ vs. $D_{\max}$ relationship given by Eq. (4). *Right:* $v$ vs. $A$ relationship given by Eq. (5). The same data are shown in Table 3 for $v$ vs. $D_{\max}$ and in Table 4 for $v$ vs. $A$.

effect on the fit. The binned data consists of the median values in each bin, i.e., median fall speeds versus median maximum dimensions and median fall speeds versus median cross-sectional areas. This method of fitting to binned data is further referred to as $M_{\text{b}}$. Previous studies have also used a similar method of binning fall speed before data fitting (e.g., Barthazy and Schefold, 2006; Zawadzki et al., 2010).


Since the binned data is based on the individual data, the fits obtained from the binned data ($M_{\text{b}}$) should match the fits based on the individual data ($M_{\text{a}}$). When they do match, and, in particular, when $R^2$ is high, the fits are deemed representative for the given shape group. If $R^2$ remains low after binning, it implies that no reliable relationship could be found. This may indicate that no adequate fit exists for that particular shape group, or it may be the consequence of too much spread in the fall speed

data obscuring any relationship. For example, Fig. 6 shows the binned data of shape group *(5) Plates* and the corresponding fit, which matches closely the fit to $M_{\text{a}}$. After binning, the correlation coefficients are much higher for both the $v$ vs. $D_{\max}$ and the $v$ vs. $A$ relationships with $R^2 \simeq 0.89$ (Table 3) and $R^2 \simeq 0.87$ (Table 4), respectively. Therefore, for this shape group, the fits $M_{\text{b}}$ can be considered representative.

The method $M_{\text{a}}$ fits agree reasonably well with $M_{\text{b}}$ fits for all shape groups if considering confidence regions (see Figures A1

and A2 in Appendix A). Therefore, considering the correlation coefficients $R^2$ from the binned data ($M_{\text{b}}$) is necessary in order to judge if the relationships are reliable or not. For clarity, $R^2$ for the two types of comparisons are denoted differently so that $R_{\text{a}}^2$ and $R_{\text{b}}^2$ are the individual correlation scores for the fits to $v$ vs. $D_{\max}$ and $v$ vs. $A$ relationships, respectively. They are



compared to each other in Fig. 7, which shows that the correlation coefficients of seven out of 15 groups improve past 0.5 in
$M_\mathrm{b}$ for both $R_\mathrm{a}^2$ and $R_\mathrm{b}^2$ (see Tables 3 and 4), including three shape groups with a very high correlation to their fit ($R^2 \gtrsim 0.9$),
namely groups *(5) Plates*, *(11) Spatial stellar*, and *(12) Graupel*. The other groups with good correlation are *(7) Bullet rosettes*,
*(8) Branches*, *(14) Irregulars*, and *(15) Spherical*. For all other eight shape groups, one or both of $R_\mathrm{a}^2$ and $R_\mathrm{b}^2$ remain below
0.5. Therefore, for these groups, no solid relationship could be found for $v$ vs. $D_\mathrm{max}$, or $v$ vs. $A$, or both.

### 3.2.3 Comparing size and cross-sectional area dependencies

For the seven groups with good correlations, $R_\mathrm{a}^2$ and $R_\mathrm{b}^2$ are similar (see also Fig. 7). As discussed in Sect. 3.1, particle size and
cross-sectional area are very well correlated, so this is expected. Only for two of the other eight groups, are the values of $R_\mathrm{a}^2$
and $R_\mathrm{b}^2$ similar. While for shape group *(4) Capped columns*, binning the data made a similar improvement to both $R_\mathrm{a}^2$ and $R_\mathrm{b}^2$,
increasing the correlation towards 0.5, for *(10) Spatial plates*, both $R_\mathrm{a}^2$ and $R_\mathrm{b}^2$ remain very low for $M_\mathrm{b}$. For the remaining six
groups, there is a noticeable difference between $R_\mathrm{a}^2$ and $R_\mathrm{b}^2$. On the one hand, shape groups *(6) Stellar* and *(13) Ice particles*
have improved their correlation coefficients $R_\mathrm{a}^2$ to above 0.5, but without an improvement in $R_\mathrm{b}^2$. On the other hand, groups
*(1) Needles*, *(2) Crossed needles*, *(3) Thick columns*, and *(9) Side planes* have $R_\mathrm{b}^2$ values that are significantly larger than
the respective $R_\mathrm{a}^2$ values. For example, shape group *(1) Needles* has $R_\mathrm{a}^2 = 0.24$ and $R_\mathrm{b}^2 = 0.50$, and *(3) Thick columns* has
$R_\mathrm{a}^2 = 0.11$ and $R_\mathrm{b}^2 = 0.44$. For the groups *(2) Crossed needles* and *(9) Side planes*, the difference between $R_\mathrm{a}^2$ and $R_\mathrm{b}^2$ is most
pronounced with no improvement in $R_\mathrm{a}^2$ but moderate values for $R_\mathrm{b}^2$ of 0.36 and 0.50, respectively.

The results discussed above show that among these groups with a noticeable difference between $R_\mathrm{a}^2$ and $R_\mathrm{b}^2$, more have larger
$R_\mathrm{b}^2$ (four groups) than larger $R_\mathrm{a}^2$ (two groups), i.e., more have better $v$ versus $A$ correlation than $v$ versus $D_\mathrm{max}$. Particles are
falling at a speed for which gravitational and drag forces are in equilibrium, i.e., fall speed depends on mass and drag, which in
turn depends on cross-sectional area $A$ and the drag coefficient $C_\mathrm{D}$ (e.g., Mitchell, 1996). Since drag depends directly on cross-
sectional area, one may expect fall speed to depend more on the cross-sectional area than on particle size. Drag, in addition
to cross-sectional area, also depends on $C_\mathrm{D}$, which is proportional to the particle Reynolds number, which in turn depends
on a characteristic length of the particle. For most particle shapes, $D_\mathrm{max}$ may be a good approximation for this characteristic
length; hence, fall speed also depends directly on $D_\mathrm{max}$. However, for some shapes, $D_\mathrm{max}$ may be significantly different from
the characteristic length for the Reynolds number, so that fall speed is not necessarily well correlated to $D_\mathrm{max}$.

For example, for needles or columns, if falling horizontally, this characteristic length is given by the needle's or column's
width rather than its maximum dimension $D_\mathrm{max}$, which is similar to the needle's or column's length. Indeed, the shape groups
related to needles and columns, i.e., *(1) Needles*, *(2) Crossed needles*, and *(3) Thick columns*, are among the four groups for
which fall speed is better correlated to $A$ than to $D_\mathrm{max}$. Interestingly, as seen in Sect. 3.1, these three shape groups also have the
lowest exponents $b$ for the $A$ vs. $D_\mathrm{max}$ relationships with values close to 1. Two of these groups, *(1)* and *(2)*, are also among the
four groups with the lowest correlation between $A$ and $D_\mathrm{max}$ (together with shape groups *(4) Capped columns* and *(6) Stellar*),
indicating again that the differences between $R_\mathrm{a}^2$ and $R_\mathrm{b}^2$ that we see in three of these four groups are not unexpected.





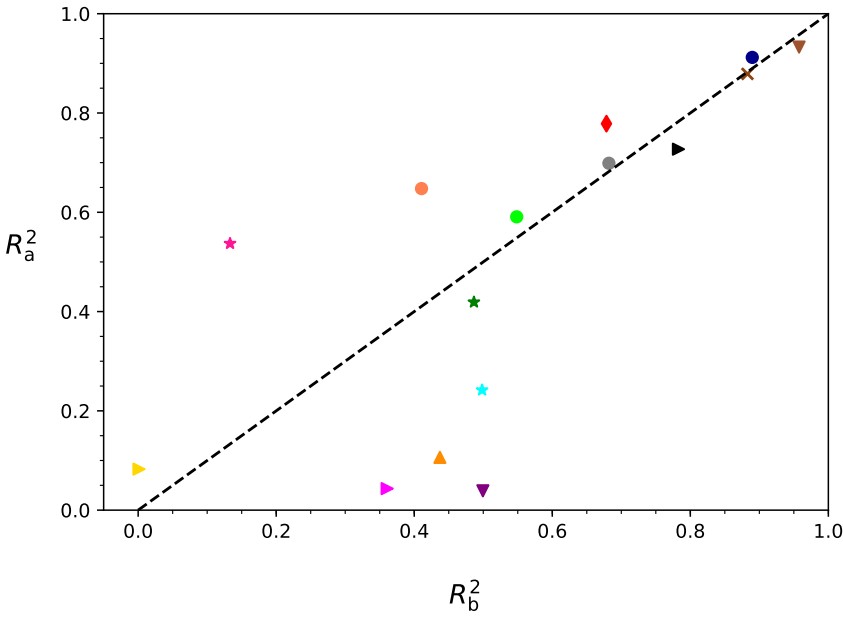

**Figure 7.** The correlation coefficients $R_a^2$ ($v$ versus $D_{max}$) and $R_b^2$ ($v$ versus $A$) from $M_b$ are shown for the 15 shape groups. For a legend of the shape groups, see Fig. 5. The black dashed line represents $R_a^2 = R_b^2$.

### 3.2.4 Representative relationships

Figure 8 shows the fitted $v$ vs. $D_{max}$ (*left*) and $v$ vs. $A$ (*right*) relationships (from method $M_b$) as solid lines for the shape groups with relatively good correlations ($R^2 \gtrsim 0.50$). These are shape groups *(5) Plates*, *(6) Stellar*, *(7) Bullet rosettes*, *(8) Branches*, *(11) Spatial stellar crystals*, *(12) Graupel*, *(13) Ice particles*, *(14) Irregulars*, and *(15) Spherical* for the $v$ vs. $D_{max}$ relationships. For the $v$ vs. $A$ relationships, the correlated shape groups are *(1) Needles*, *(4) Capped columns*, *(5) Plates*, *(7) Bullet rosettes*, *(8) Branches*, *(9) Side planes*, *(11) Spatial stellar*, *(12) Graupel*, *(14) Irregulars*, and *(15) Spherical*. For comparison, the relationships for all shapes combined appear as dashed lines in both figures.

Given by their compact shape, *(15) Spherical* have the largest mass and least drag for a given size. Therefore, they will fall faster than any other shape and have the highest slopes, i.e., values for steepest slopes, i.e., highest values of exponents $b_D$ and $b_A$. Among all shapes, *(12) Graupel* is most similar to spherical particles as they often have spheroidal shape. However, their $b_D$ (1.0) and $b_A$ (0.47) are considerably smaller than those of spheres, though still larger than for any other shape (see Fig. 8 and Tables 3-4).

While two shape groups (*(6) Stellar* and *(11) Spatial stellar*) have similarly large values of $b_D$, the relationships shift towards much lower speeds and larger sizes compared to the relationships of shape groups *(12) Graupel* and *(15) Spherical* (see Fig. 8). Shape group *(11)* also has a similarly large value of $b_A$ as shape group *(12)*, but again its relationship shifts towards lower speeds and this time larger cross-sectional areas. The other groups with $R_a^2 > 0.5$ have $b_D$ values around 0.4, and the other





**Table 3.** Fall speed versus particle size ($v$ vs. $D_{\mathrm{max}}$) relationships fitted to Eq. (4) for each shape group and all data, i.e., for all the particles regardless of shape. The parameters $a_D$, $b_D$ with their respective uncertainties, and the correlation coefficients ($R_{\mathrm{a}}^2$ and $R_{\mathrm{b}}^2$) are shown for both methods ($M_{\mathrm{a}}$ and $M_{\mathrm{b}}$) for each shape group and regardless of shape.

| | $v$ vs. $D_{\mathrm{max}}$ | | | | | |
| | Method $M_{\mathrm{a}}$ | | | Method $M_{\mathrm{b}}$ | | |
| Shape groups (1–15) | $a_D$ [m s$^{-1}$] | $b_D$ | $R_{\mathrm{a}}^2$ | $a_D$ [m s$^{-1}$] | $b_D$ | $R_{\mathrm{b}}^2$ |
|---|---|---|---|---|---|---|
| *(1) Needles* | $0.34 \pm 0.02$ | $-0.03 \pm 0.05$ | 0.001 | $0.35 \pm 0.05$ | $-0.15 \pm 0.11$ | 0.24 |
| *(2) Crossed needles* | $0.35 \pm 0.06$ | $0.01 \pm 0.13$ | 0.0002 | $0.35 \pm 0.05$ | $-0.07 \pm 0.13$ | 0.04 |
| *(3) Thick columns* | $0.36 \pm 0.08$ | $0.19 \pm 0.09$ | 0.05 | $0.34 \pm 0.14$ | $0.12 \pm 0.15$ | 0.11 |
| *(4) Capped columns* | $0.48 \pm 0.03$ | $0.33 \pm 0.06$ | 0.12 | $0.49 \pm 0.07$ | $0.28 \pm 0.14$ | 0.42 |
| *(5) Plates* | $0.50 \pm 0.04$ | $0.40 \pm 0.05$ | 0.24 | $0.51 \pm 0.05$ | $0.39 \pm 0.06$ | 0.88 |
| *(6) Stellar* | $0.26 \pm 0.10$ | $0.67 \pm 0.21$ | 0.20 | $0.23 \pm 0.17$ | $0.99 \pm 0.37$ | 0.54 |
| *(7) Bullet rosettes* | $0.59 \pm 0.04$ | $0.51 \pm 0.14$ | 0.26 | $0.62 \pm 0.05$ | $0.44 \pm 0.15$ | 0.59 |
| *(8) Branches* | $0.34 \pm 0.02$ | $0.33 \pm 0.06$ | 0.07 | $0.35 \pm 0.03$ | $0.36 \pm 0.08$ | 0.78 |
| *(9) Side planes* | $0.52 \pm 0.02$ | $0.14 \pm 0.05$ | 0.02 | $0.54 \pm 0.02$ | $0.04 \pm 0.08$ | 0.04 |
| *(10) Spatial plates* | $0.44 \pm 0.09$ | $0.21 \pm 0.19$ | 0.03 | $0.44 \pm 0.13$ | $0.20 \pm 0.28$ | 0.08 |
| *(11) Spatial stellar* | $0.46 \pm 0.03$ | $0.70 \pm 0.05$ | 0.48 | $0.45 \pm 0.04$ | $0.88 \pm 0.10$ | 0.93 |
| *(12) Graupel* | $0.98 \pm 0.08$ | $0.89 \pm 0.11$ | 0.65 | $1.07 \pm 0.09$ | $1.00 \pm 0.13$ | 0.91 |
| *(13) Ice particles* | $0.61 \pm 0.12$ | $0.38 \pm 0.10$ | 0.21 | $0.65 \pm 0.14$ | $0.39 \pm 0.12$ | 0.65 |
| *(14) Irregulars* | $0.44 \pm 0.03$ | $0.37 \pm 0.05$ | 0.16 | $0.46 \pm 0.07$ | $0.46 \pm 0.12$ | 0.70 |
| *(15) Spherical* | $4.49 \pm 0.28$ | $1.37 \pm 0.16$ | 0.67 | $4.76 \pm 0.63$ | $1.42 \pm 0.35$ | 0.73 |
| **All data** | $0.42 \pm 0.01$ | $0.20 \pm 0.02$ | 0.06 | $0.44 \pm 0.02$ | $0.19 \pm 0.03$ | 0.87 |

groups with $R_{\mathrm{b}}^2 > 0.5$ have $b_A$ values around 0.2 (0.16 to 0.27) except for shape group *(9) Side planes*, which has the smallest value (0.11).

There seems to be around factor 2 between $b_A$ and $b_D$. By combining Equations 4, 5, and 2 one finds that $b$ should give this factor. As can be seen in Table 1, the coefficient $b$ is for most shape groups between 1.5 and 2. Figure 9 shows the ratios $\frac{b_D}{b_A}$ as a function of $b$, and most ratios on this plot are close to the line $\frac{b_D}{b_A} = b$. The exceptions are the two shape groups where $R_{\mathrm{a}}^2$ was larger than $R_{\mathrm{b}}^2$ (*(6) Stellar*, and *(10) Spatial plates*), which are found above the line. Group *(10)* is outside the plot domain since it has an excessively high ratio of 9.37 caused by a very small $b_D$. However, this is probably not meaningful since the correlation is very bad for this group. The four shape groups with $R_{\mathrm{b}}^2$ larger than $R_{\mathrm{a}}^2$ (*(1) Needles*, *(2) Crossed needles*, *(3) Thick columns*, and *(9) Side planes*), are below the line.



**Table 4.** Fall speed versus cross-sectional area ($v$ vs. $A$) relationships fitted to Eq. (5) for each shape group and for all data, i.e., for all the particles regardless of shape. The parameters $a_A$, $b_A$ with their respective uncertainties, and the correlation coefficients ($R_a^2$ and $R_b^2$) are shown for both methods ($M_a$ and $M_b$) for each shape group and regardless of shape.

| | $v$ vs. $A$ | | | | | |
| --- | --- | --- | --- | --- | --- | --- |
| | **Method $M_a$** | | | **Method $M_b$** | | |
| **Shape groups (1–15)** | $a_A$ [m s$^{-1}$] | $b_A$ | $R_a^2$ | $a_A$ [m s$^{-1}$] | $b_A$ | $R_b^2$ |
| *(1) Needles* | $0.51 \pm 0.07$ | $0.21 \pm 0.04$ | $0.10$ | $0.50 \pm 0.16$ | $0.20 \pm 0.08$ | $0.50$ |
| *(2) Crossed needles* | $0.54 \pm 0.15$ | $0.30 \pm 0.10$ | $0.12$ | $0.57 \pm 0.27$ | $0.33 \pm 0.18$ | $0.36$ |
| *(3) Thick columns* | $0.73 \pm 0.17$ | $0.31 \pm 0.06$ | $0.22$ | $0.60 \pm 0.34$ | $0.26 \pm 0.12$ | $0.44$ |
| *(4) Capped columns* | $0.57 \pm 0.07$ | $0.17 \pm 0.04$ | $0.10$ | $0.60 \pm 0.12$ | $0.16 \pm 0.07$ | $0.49$ |
| *(5) Plates* | $0.59 \pm 0.06$ | $0.22 \pm 0.03$ | $0.23$ | $0.57 \pm 0.06$ | $0.20 \pm 0.03$ | $0.88$ |
| *(6) Stellar* | $0.37 \pm 0.08$ | $0.31 \pm 0.12$ | $0.14$ | $0.37 \pm 0.14$ | $0.24 \pm 0.25$ | $0.13$ |
| *(7) Bullet rosettes* | $0.81 \pm 0.10$ | $0.30 \pm 0.08$ | $0.28$ | $0.79 \pm 0.11$ | $0.24 \pm 0.09$ | $0.55$ |
| *(8) Branches* | $0.43 \pm 0.04$ | $0.20 \pm 0.03$ | $0.09$ | $0.45 \pm 0.07$ | $0.20 \pm 0.06$ | $0.68$ |
| *(9) Side planes* | $0.57 \pm 0.03$ | $0.11 \pm 0.03$ | $0.05$ | $0.59 \pm 0.04$ | $0.11 \pm 0.05$ | $0.50$ |
| *(10) Spatial plates* | $0.43 \pm 0.17$ | $0.04 \pm 0.11$ | $0.004$ | $0.40 \pm 0.40$ | $0.02 \pm 0.25$ | $0.001$ |
| *(11) Spatial stellar* | $0.67 \pm 0.04$ | $0.40 \pm 0.03$ | $0.51$ | $0.70 \pm 0.05$ | $0.47 \pm 0.04$ | $0.96$ |
| *(12) Graupel* | $1.35 \pm 0.11$ | $0.46 \pm 0.05$ | $0.69$ | $1.40 \pm 0.14$ | $0.47 \pm 0.07$ | $0.89$ |
| *(13) Ice particles* | $1.01 \pm 0.19$ | $0.30 \pm 0.06$ | $0.31$ | $0.87 \pm 0.38$ | $0.24 \pm 0.12$ | $0.41$ |
| *(14) Irregulars* | $0.56 \pm 0.04$ | $0.23 \pm 0.03$ | $0.19$ | $0.60 \pm 0.12$ | $0.27 \pm 0.08$ | $0.68$ |
| *(15) Spherical* | $5.42 \pm 0.29$ | $0.68 \pm 0.08$ | $0.69$ | $5.92 \pm 0.59$ | $0.71 \pm 0.15$ | $0.78$ |
| **All data** | $0.52 \pm 0.02$ | $0.18 \pm 0.01$ | $0.14$ | $0.54 \pm 0.02$ | $0.18 \pm 0.01$ | $0.97$ |

## 3.3 Particle orientation and area ratio

### 3.3.1 Orientation

For certain shapes, the orientation of the falling particle can considerably change the cross-sectional area seen in the top-view image. Therefore, the particle orientation will influence the drag and thus the fall speed. To test how much this affects our data, particles that clearly show a horizontal or vertical orientation are selected among predominantly elongated particles, found within the shape group *(1) Needles*, or predominantly planar particles found within one of the two groups *(5) Plates* and *(6) Stellar*. Particles that are identified by eye as having an orientation angle close to 0° are considered horizontal, and conversely, particles with an orientation angle close to 90° are considered vertical. The orientation angle is here defined as the angle that the horizontal plane forms with the particle plane, in case of planar particles, or with the particle axis, in case of elongated particles. Only a total of 135 particles fulfilled these criteria, 109 with horizontal and 26 with vertical orientation. Figure 10





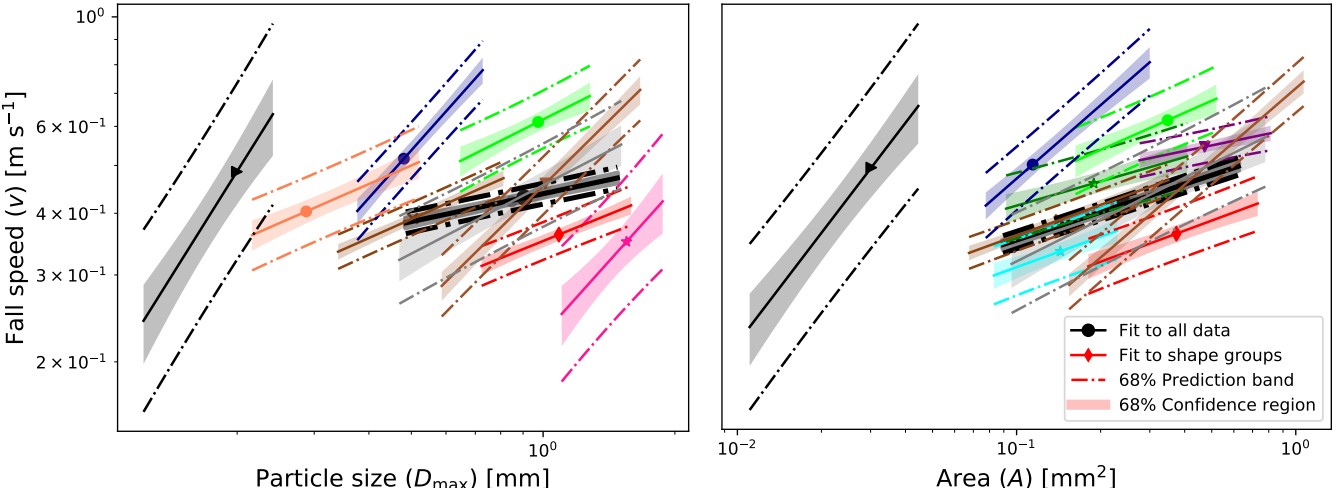

**Figure 8.** Fall speed versus particle size ($v$ vs. $D_{\mathrm{max}}$) and fall speed versus cross-sectional area ($v$ vs. $A$) relationships for the shape groups where we have found a good correlation (solid lines) and all data regardless of shape (black dashed lines) are shown. The 68% prediction band and the 68% confidence region for the fits ($M_{\mathrm{b}}$). For a legend of the shape groups, see Fig. 5. *Left: $v$ vs. $D_{\mathrm{max}}$ relationship. Shape groups (5), (6), (7), (8), (11), (12), (13), (14) and (15)* are displayed. The values of the median of $D_{\mathrm{max}}$ are represented as points. *Right: $v$ vs. $A$ relationship. Shape groups (1), (4), (5), (7), (8), (9), (11), (12), (14), and (15)* are displayed. The values of the median of $A$ are represented as points. The length of the fit lines is defined by 16th and 84th percentiles of $D_{\mathrm{max}}$ (*left*) and $A$ (*right*). The corresponding data are shown in Tables 3-4.

shows six examples (a–f) using side-view images of individual particles with horizontal orientation and six examples (g–l) with vertical orientations.

Figure 11 shows the individual fall speeds of these particles. When trying to fit these data to Eq. (4) or 5, the correlation coefficients remained very low, and thus no meaningful relationships could be found. However, particles falling with a vertical orientation are slightly faster (with a median $v = 0.42\,\mathrm{m\,s^{-1}}$) than the horizontally orientated (with a median $v = 0.34\,\mathrm{m\,s^{-1}}$).

### 3.3.2    Area ratio

In addition to orientation, also area ratio, $A_{\mathrm{r}}$, may be important, especially given that the Reynolds number, which influences fall speed (Sect. 3.2), can be related, in part, to the area ratio (Heymsfield and Westbrook, 2010). In general, the smallest particles tend to have the largest $A_{\mathrm{r}}$, and $A_{\mathrm{r}}$ becomes smaller for larger particles. This is true for most shape groups, and this tendency is particularly strong in the four shape groups *(1) Needles, (2) Crossed needles, (3) Thick columns*, and *(13) Ice particles* (see

Fig. 4), of which groups *(1)–(3)* are elongated shapes. The lowest $A_{\mathrm{r}}$, at any given size, are found in these shape groups. The elongated shapes also showed a particular size dependence of their cross-sectional area (Sect. 3.1). This dependence of area ratio and of cross-sectional area on particle size leads to a particular fall speed behaviour, which can be better visualized by splitting the data into different $A_{\mathrm{r}}$ ranges. Figure 12 shows this after splitting the data equally into three distinct regions of




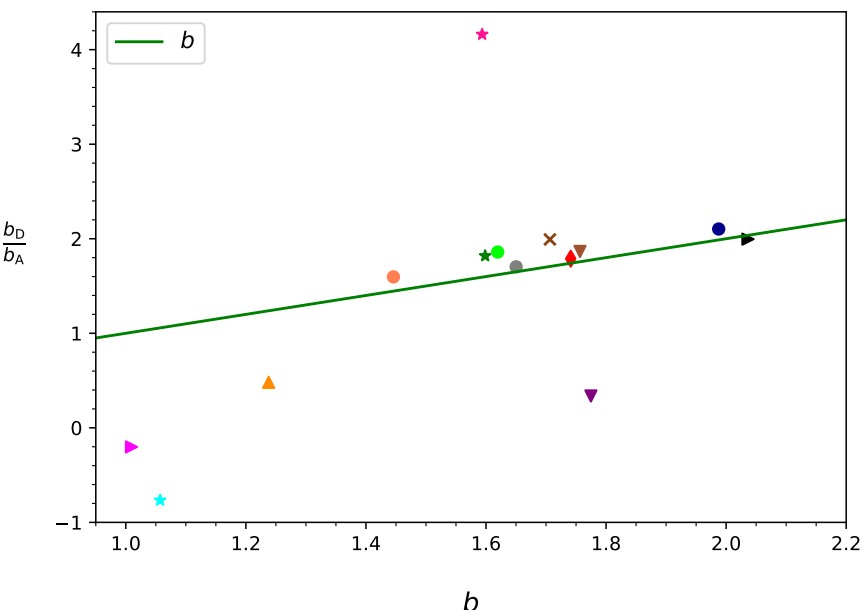

**Figure 9.** Ratio of the coefficients $b_D$ and $b_A$ from fits ($M_b$) to $v$ vs. $D_{\max}$ and $v$ vs. $A$ relationships, respectively, and the coefficient $b$ corresponding to $A$ vs. $D_{\max}$ relationship are shown for all the shape groups. The green solid line corresponds to $\frac{b_D}{b_A} = b$.

low, intermediate, and high $A_r$ values. In each range, there is a different fall speed relationship for both $v$ vs. $D_{\max}$ and $v$ vs. $A$. These relationships are overlapping somewhat in $D_{\max}$ and $A$, respectively. Figure 12 illustrates that for a given size or cross-sectional area, higher and lower $A_r$ means higher and lower fall speed, respectively. One may expect the effects of orientation to be responsible since the same elongated particle would have a relatively larger $A_r$ when oriented vertically, and thus falling faster, compared to when oriented horizontally. However, a closer inspection of the data shows that the majority of particles are horizontally oriented. The predominance of the horizontal orientation is probably a consequence of vertically falling particles being less aerodynamically stable and, thus, likely to transition to horizontal orientation. Therefore, particle orientation does not appear to explain the dependence of fall speed on area ratio. Instead, particles with higher area ratios are generally bulkier, i.e., needles or columns that are shorter in length, and consequently fall faster.

To better understand this area ratio dependence of fall speed, we first consider elongated particles that have the same $D_{\max}$ (approximately given by the length) but different values of $A_r$. Note, that the top-view images, used to determine $A$, always present a view perpendicular to the major axis of elongated particles, if these are horizontally oriented. Therefore, the cross-sectional area is approximately given by the length of the particles times its diameter $d$ (with diameter we refer to the width perpendicular to the major axis), i.e., $A \approx D_{\max} \cdot d$.

Also, as can be seen in Eq. (1), the cross-sectional area $A$ is proportional to $A_r$ for the case of $D_{\max} = $ constant as considered here. Then, also their diameter is proportional to $A_r$. Consequently, their volume ($\approx D_{\max} \cdot d^2$) or mass ($m$) is proportional to $A_r^2$. While $A \propto A_r$ is valid in general for all shapes, the strong dependence $m \propto A_r^2$ is distinctive for elongated shapes. Then,



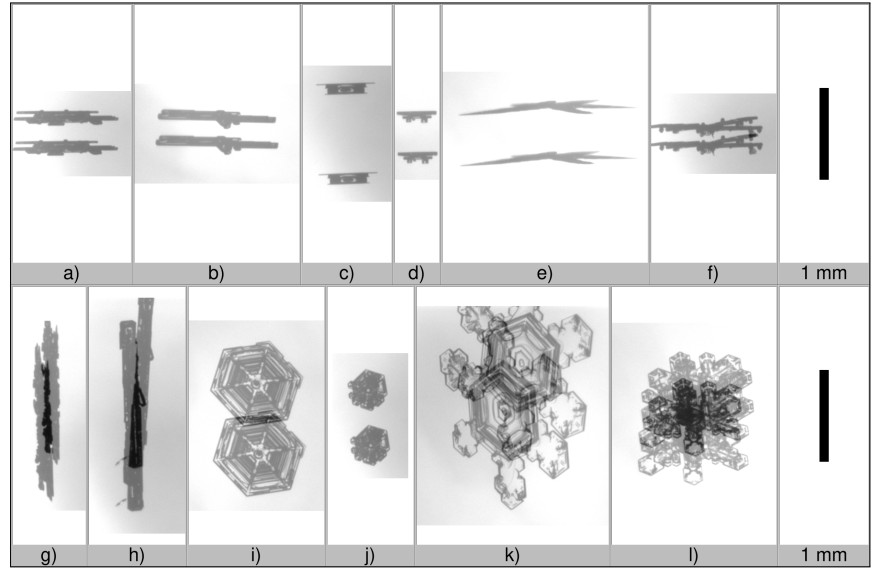

**Figure 10.** Panels **(a–f)** show six examples using side views of different particles with horizontal orientation and panels **(g–l)** show six examples using side views of different particles with vertical orientation. These particles have elongated shape (shape group *(1) Needles*) and planar shape (shape groups *(5) Plates* and *(6) Stellar*). Two examples of each shape group are displayed for both orientations. The same scaling is applied to all images; a 1 mm scale bar is shown as reference.

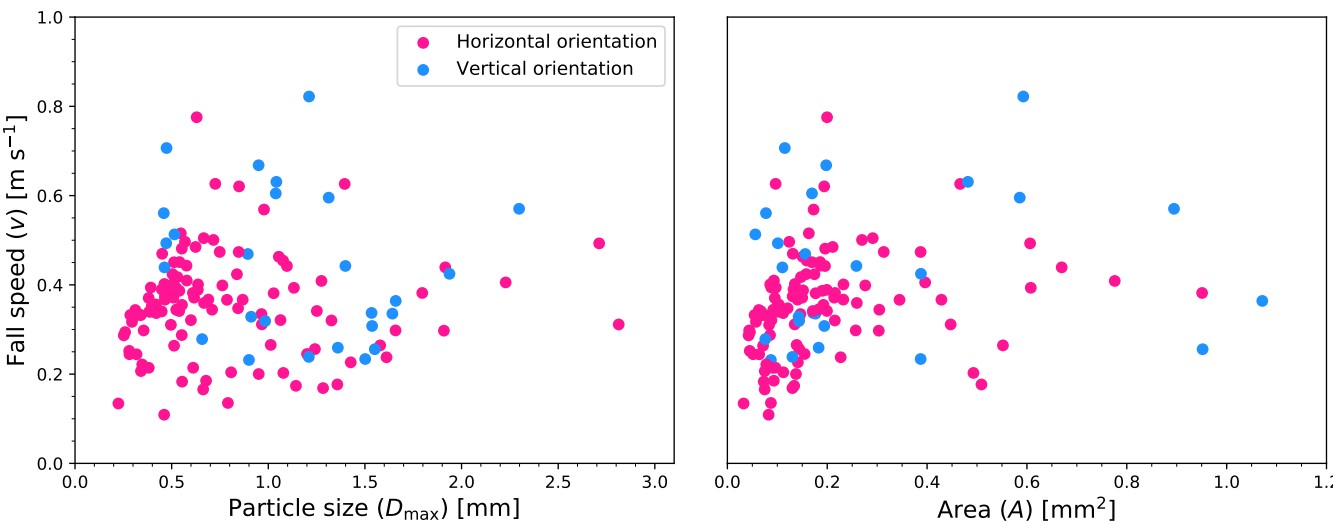

**Figure 11.** Fall speed versus particle size ($v$ vs. $D_{max}$) and fall speed versus cross-sectional area ($v$ vs. $A$) for mixed particle shapes, predominantly elongated particles (shape group *(1) Needles*) and planar particles (shape groups *(5) Plates* and *(6) Stellar*) with horizontal and vertical orientation angles. *Left:* $v$ vs. $D_{max}$ relationship. *Right:* $v$ vs. $A$.





**Table 5.** Fall speed versus particle size ($v$ vs. $D_{\max}$) and fall speed versus cross-sectional area ($v$ vs. $A$) relationships fitted to binned data ($M_b$) for the combination of particles in shape groups *(1) Needles*, *(2) Crossed needles*, and *(3) Thick columns* with different area ratio $A_r$ ranges (low, intermediate and high). The number of particles, $N$, the parameters $a_D$, $b_D$, $a_A$, $b_A$ and their respective uncertainties, and the correlation coefficients $R_b^2$ are shown. All data are also shown. In this case, all data include only particles in these three shape groups, i.e., *(1)–(3)*.

| Ranges | $N$ | $v$ vs. $D_{\max}$ | | | $v$ vs. $A$ | | |
| --- | --- | --- | --- | --- | --- | --- | --- |
| | | $a_D$ [m s$^{-1}$] | $b_D$ | $R_b^2$ | $a_A$ [m s$^{-1}$] | $b_A$ | $R_b^2$ |
| All data (1–3) | 486 | $0.34 \pm 0.04$ | $0.01 \pm 0.07$ | 0.01 | $0.50 \pm 0.10$ | $0.20 \pm 0.05$ | 0.74 |
| $A_r$ low | 161 | $0.23 \pm 0.05$ | $0.64 \pm 0.11$ | 0.84 | $0.58 \pm 0.08$ | $0.43 \pm 0.05$ | 0.93 |
| $A_r$ intermediate | 164 | $0.37 \pm 0.04$ | $0.72 \pm 0.14$ | 0.82 | $0.87 \pm 0.12$ | $0.48 \pm 0.06$ | 0.91 |
| $A_r$ high | 161 | $0.50 \pm 0.07$ | $0.38 \pm 0.08$ | 0.80 | $0.93 \pm 0.20$ | $0.35 \pm 0.08$ | 0.78 |

for these shapes, as $A_r$ increases, mass increases much more rapidly than $A$, and consequently, fall speed needs to increase considerably for drag to compensate gravitational force. This effect can explain the strong dependence of fall speed on area ratio for these elongated shapes.

For other shapes, the general dependence may be similar, though less pronounced due to a weaker $A_r$ dependence of $m$.
Additionally, for these other shapes, the range of $A_r$ is not as wide as for the elongated shapes. Moreover, for no other shape group do the fall speeds separate into distinguishable relationships after splitting the data according to $A_r$. That indicates that the natural spread in fall speed may hide the $A_r$ dependence of fall speed.

To examine further, we also consider what happens at increasing $D_{\max}$ in the case of constant $A_r$. In this case, $A \propto D_{\max}^2$, and $m \propto D_{\max}^3$ in general. Consequently, as $m$ increases much more rapidly with increasing $D_{\max}$ than $A$, the fall speed also
increases rapidly with increasing $D_{\max}$, which is consistent with the strongest size dependence of fall speed existing in shape groups *(12) Graupel* and *(15) Spherical* (see Sect. 3.2).

Finally, considering the general case when neither $D_{\max}$ nor $A_r$ are constant, one needs to take into account both of the special cases explained above. On the one hand, increasing $D_{\max}$ leads directly to increasing fall speed. On the other hand, increasing $D_{\max}$ changes the particle morphology so that $A_r$ decreases, which, in turn, causes fall speed to decrease. Since
these effects are opposed, they cancel each other out to some extent. The stronger the negative size dependence of $A_r$ is, the weaker the positive size dependency of fall speed. If the effect related to $A_r$ is the strongest, they cancel out almost entirely, as in case of shape groups *(1)–(3)* where this results in the weakest size dependence of fall speed with low correlation. Another consequence of the $A_r$ dependence of fall speed is that variations in $A_r$ cause variations in fall speed, i.e., they account in part for the natural spread in the data.





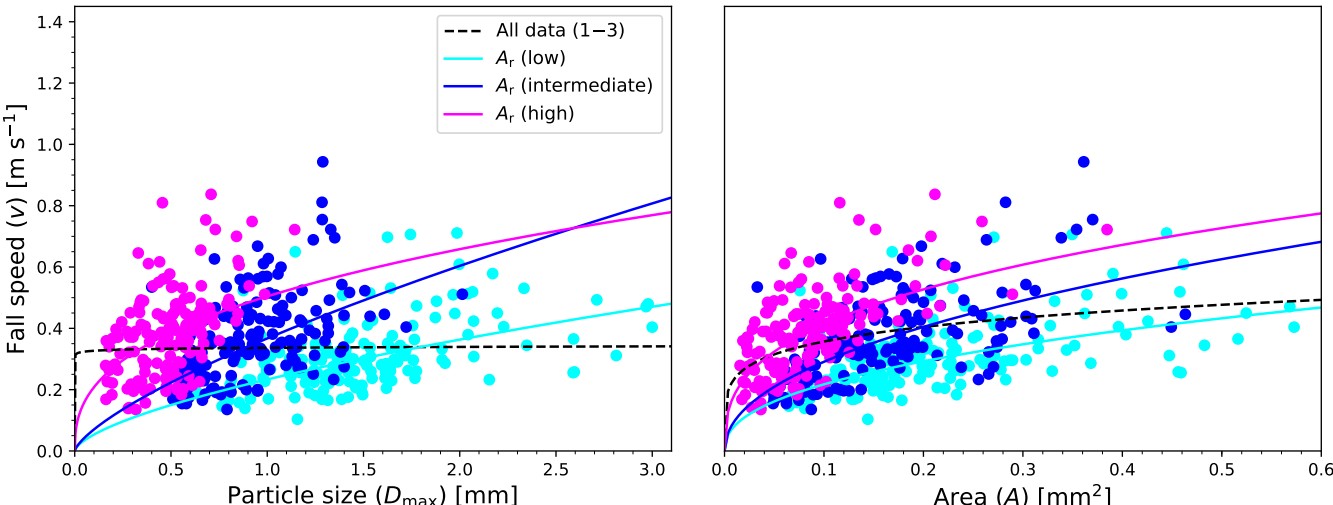

**Figure 12.** Fall speed versus particle size *(left)* and fall speed versus cross-sectional area *(right)* relationships for the combination of shape groups *(1) Needles*, *(2) Crossed needles*, and *(3) Thick columns*. Fits that apply to binned data ($M_b$) are shown for all the data (solid black line) and for different $A_r$ (low, intermediate and high) ranges. All data, in this case, only include particles in these three shape groups, i.e., *(1)–(3)*. For the binned data, the median fall speed in the size and cross-sectional area bins, and the median was chosen. Same data are shown in Table 5.

### 3.4 Comparison with previous fall speed relationships

This section compares the fall speed relationships as functions of particle size presented in this study, further referred to as *[VM]*, to parameterizations of previous studies. Here, we assess the shape groups *(5) Plates*, *(6) Stellar* (called dendrites in other studies), and *(12) Graupel*, where the data are reasonably well correlated with the fall speed relationships ($R^2 \gtrsim 0.5$). Table 6 lists the parameterizations of Locatelli and Hobbs (1974) *[Lo]*, Heymsfield and Kajikawa (1987) *[H]*, Mitchell (1996) *[M]*, Barthazy and Schefold (2006) *[B]*, and Lee et al. (2015) *[Le]* used in the comparison.

Before comparing, it is important to note that the particle size $D$ was defined somewhat differently depending on the study. For *[VM]*, as well as for *[H]* and *[M]*, $D$ corresponds to $D_{max}$. For *[Lo]*, $D$ is the diameter of an estimated circle that has the same cross-sectional area as the imaged particle, and for *[B]* and *[Le]*, $D$ corresponds to the maximum length of any horizontal row in the side-view shadowgraphs. *[Lo]* studied fall speeds of different types of ice crystals by first measuring the fall speed of individual particles and then subsequently collecting and imaging them. Their fitted relationships of fall speed are often used as a reference by other studies in the literature. *[H]* also used data from fall speed measurements and subsequent imaging of individual snow particles, which were collected by Kajikawa (1972). What *[Lo]* and *[H]* have in common with *[VM]* is that all ice particles that contribute to a fall speed parameterization are individually shape classified and therefore belong to the studied shape. *[B]*, on the other hand, loosely tied particle shape to fall speeds by determining the dominant particle shape (occurrence $> 50\%$) per time interval from an independent instrument, and later associated the fall speeds in the time interval





to the dominant shape. *[Le]* used a method similar to *[B]*; however, they used a higher occurrence threshold of 70%. The fall speed parameterizations of the study by *[M]* are predicted from previous literature relationships of cross-sectional area and mass versus particle size.

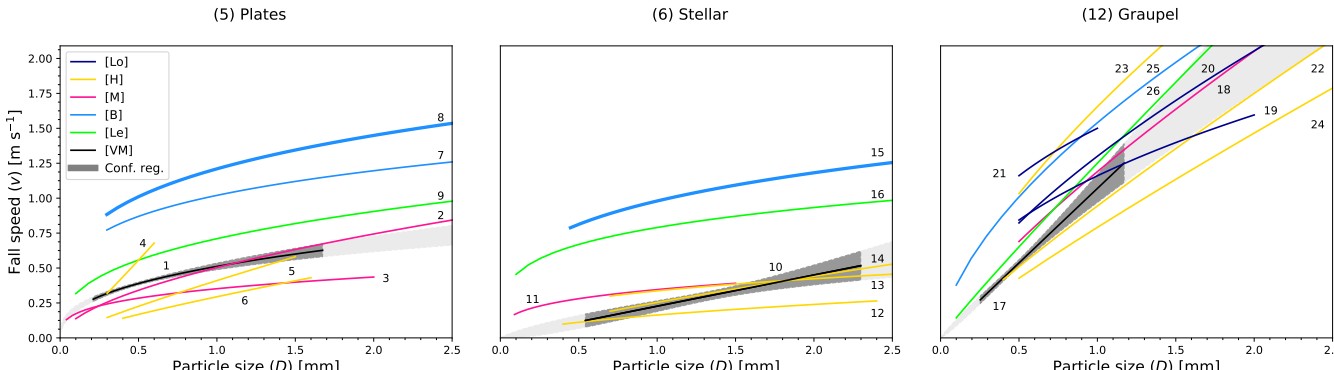

**Figure 13.** A comparison of the fall speed versus particle size ($v$ vs. $D$) relationships between this study and previous studies for some shape groups: *(5) Plates*, *(6) Stellar*, and *(12) Graupel*. For the comparison, $v$ vs. $D$ parameterizations from Locatelli and Hobbs (1974) *[Lo]*, Heymsfield and Kajikawa (1987) *[H]*, Mitchell (1996) *[M]*, Barthazy and Schefold (2006) *[B]*, Lee et al. (2015) *[Le]*, and this work *[VM]* are shown. These $v$ vs. $D$ relationships are the same shown and enumerated in Table 6. The thickness in lines corresponding to *[B]* represent the riming degree, the thinner line denotes 'unrimed', the thicker denotes 'moderately rimed'. The power laws that correspond to *[VM]* are shown together with their respective 68% confidence regions ($M_b$). The lengths of all relationships correspond to the ranges of $D$ (see Table 6).

Figure 13 shows that, for plates, the previous relationships by *[H]* and *[M]* are closest to results from *[VM]*. While their

relationships for crystal with sector-like branches (P1b) produce slower fall speeds than *[VM]*, their relationships for plates are closer and extend into or cross the confidence region of *[VM]*. *[H]* also reported a relationship for thick plates, which in most of its size range, is just above *[VM]*. The relationships reported by *[B]* have the highest fall speeds for plates. They reported different relationships for different degrees of riming, with more riming leading to higher fall speeds. Our data included in shape group *(5) Plates* are unrimed. However, even the unrimed plates from *[B]* appear to be much faster. The relationships

from *[B]* may overestimate fall speeds because of their classification method mentioned above, which allowed up to half of the particles that contributed to the relationship to have different shapes. *[Le]* also reported relationships for plates' speed that are faster than our relationship, although much closer and considerably slower than *[B]*. The better agreement is possibly due to a more accurate shape classification, while otherwise using a similar method to *[B]*.

As for plates, also for stellar particles the previous relationships by *[H]* and *[M]* are closest to *[VM]*. Again, *[Le]* and *[B]*

reported relationships with considerably higher fall speeds.

*[Lo]* reported three relationships for lump graupel with different densities. The higher the density, i.e., the more compact the graupel particles are, the faster their predicted fall speeds will be. The relationship for lump graupel by *[M]* bases on the mass relationship of the medium density graupel by *[Lo]*; consequently, it is very close to the corresponding fall speed relationship.





**Table 6.** The $v$ vs. $D$ relationships of previous studies given by Locatelli and Hobbs (1974) *[Lo]*, Heymsfield and Kajikawa (1987) *[H]*, Mitchell (1996) *[M]*, Barthazy and Schefold (2006) *[B]*, and Lee et al. (2015) *[Le]* are shown for some shapes that were selected for the comparison and correspond to *(5) Plates*, *(6) Stellar*, and *(12) Graupel*. The power laws for *[M]* have been determined by using equations [20] and [22] in Mitchell (1996). The relationships found in this work are also shown as *[VM]*. The snow particles type, the total number of particles $N$, ranges of particle sizes $D$, $v$ vs. $D$ relationships, the correlation coefficient $R^2$, and the reference of the studies are displayed. In some of these studies, the particle size is defined somewhat differently. However, in *[H]* and *[M]* $D$ is defined as $D_{\mathrm{max}}$ as in *[VM]*. Magono and Lee (1966)'s symbols are sometimes added for shape clarification. These $v$ vs. $D$ relationships are shown in Fig. 13.

| Snow particle type | $N$ | Range of $D$ | Relationship ($v$–$D$) | $R^2$ | Ref. |
|---|---|---|---|---|---|
| **Shape group *(5) Plates*** | 197 | 0.21–1.7 mm | 1. $v/(\mathrm{m\,s^{-1}}) = 0.51 \cdot (D/\mathrm{mm})^{0.39}$ | 0.88 | *[VM]* |
| Hexagonal plates | – | 100–3000 μm | 2. $v/(\mathrm{cm\,s^{-1}}) = 184 \cdot (D/\mathrm{cm})^{0.56}$ | – | *[M]* |
| Crystal with sector-like branches (P1b) | – | 40–2000 μm | 3. $v/(\mathrm{cm\,s^{-1}}) = 71 \cdot (D/\mathrm{cm})^{0.31}$ | – | *[M]* |
| Thick plate (C1h) | 19 | 0.30–0.6 mm | 4. $v/(\mathrm{cm\,s^{-1}}) = 1457 \cdot (D/\mathrm{cm})^{1.09}$ | 0.46 | *[H]* |
| Hexagonal plate (P1a) | 34 | 0.30–1.5 mm | 5. $v/(\mathrm{cm\,s^{-1}}) = 297 \cdot (D/\mathrm{cm})^{0.86}$ | 0.69 | *[H]* |
| Crystal with sector-like branches (P1b) | 19 | 0.40–1.6 mm | 6. $v/(\mathrm{cm\,s^{-1}}) = 190 \cdot (D/\mathrm{cm})^{0.81}$ | 0.96 | *[H]* |
| Unrimed plates | – | 0.30–2.7 mm | 7. $v/(\mathrm{m\,s^{-1}}) = 1.02 \cdot (D/\mathrm{mm})^{0.23}$ | 0.87 | *[B]* |
| Moderately rimed plates | – | 0.30–3.6 mm | 8. $v/(\mathrm{m\,s^{-1}}) = 1.21 \cdot (D/\mathrm{mm})^{0.26}$ | 0.73 | *[B]* |
| Plate | – | 0–4.0 mm | 9. $v/(\mathrm{m\,s^{-1}}) = 0.71 \cdot (D/\mathrm{mm})^{0.35}$ | – | *[Le]* |
| **Shape group *(6) Stellar*** | 43 | 0.54–2.3 mm | 10. $v/(\mathrm{m\,s^{-1}}) = 0.23 \cdot (D/\mathrm{mm})^{0.99}$ | 0.54 | *[VM]* |
| Stellar crystal with broad arms (P1d) | – | 90–1500 μm | 11. $v/(\mathrm{cm\,s^{-1}}) = 69 \cdot (D/\mathrm{cm})^{0.30}$ | – | *[M]* |
| Stellar crystal with broad arms (P1d) | 23 | 0.40–2.4 mm | 12. $v/(\mathrm{cm\,s^{-1}}) = 58 \cdot (D/\mathrm{cm})^{0.55}$ | 0.82 | *[H]* |
| Stellar with end plates (P2a) | 11 | 0.70–3.0 mm | 13. $v/(\mathrm{cm\,s^{-1}}) = 72 \cdot (D/\mathrm{cm})^{0.33}$ | 0.54 | *[H]* |
| Plate with dendritic extensions (P2g) | 10 | 0.70–2.8 mm | 14. $v/(\mathrm{cm\,s^{-1}}) = 160 \cdot (D/\mathrm{cm})^{0.80}$ | 0.89 | *[H]* |
| Moderately rimed dendrites | – | 0.45–3.7 mm | 15. $v/(\mathrm{m\,s^{-1}}) = 0.98 \cdot (D/\mathrm{mm})^{0.27}$ | 0.69 | *[B]* |
| Dendrite | – | 0–4.0 mm | 16. $v/(\mathrm{m\,s^{-1}}) = 0.79 \cdot (D/\mathrm{mm})^{0.24}$ | – | *[Le]* |
| **Shape group *(12) Graupel*** | 37 | 0.25–1.2 mm | 17. $v/(\mathrm{m\,s^{-1}}) = 1.07 \cdot (D/\mathrm{mm})^{1.00}$ | 0.91 | *[VM]* |
| Lump graupel (R4b) | – | 500–3000 μm | 18. $v/(\mathrm{cm\,s^{-1}}) = 728 \cdot (D/\mathrm{cm})^{0.79}$ | – | *[M]* |
| Lump graupel (R4b) | 35 | 0.5–2.0 mm | 19. $v/(\mathrm{m\,s^{-1}}) = 1.16 \cdot (D/\mathrm{mm})^{0.46}$ | $r = 0.55$ | *[Lo]* |
| Lump graupel (R4b) | 58 | 0.5–3.0 mm | 20. $v/(\mathrm{m\,s^{-1}}) = 1.3 \cdot (D/\mathrm{mm})^{0.66}$ | $r = 0.77$ | *[Lo]* |
| Lump graupel (R4b) | 17 | 0.5–1.0 mm | 21. $v/(\mathrm{m\,s^{-1}}) = 1.5 \cdot (D/\mathrm{mm})^{0.37}$ | $r = 0.58$ | *[Lo]* |
| Lump graupel (R4b) | 116 | 0.4–9.0 mm | 22. $v/(\mathrm{cm\,s^{-1}}) = 733 \cdot (D/\mathrm{cm})^{0.89}$ | 0.78 | *[H]* |
| R4b, $T \geq 0.5\,°\mathrm{C}$ | 31 | 0.5–4.7 mm | 23. $v/(\mathrm{cm\,s^{-1}}) = 792 \cdot (D/\mathrm{cm})^{0.68}$ | 0.92 | *[H]* |
| R4b, $T < 0.5\,°\mathrm{C}$ | 85 | 0.5–9.0 mm | 24. $v/(\mathrm{cm\,s^{-1}}) = 614 \cdot (D/\mathrm{cm})^{0.89}$ | 0.92 | *[H]* |
| Graupel | – | 0–4.8 mm | 25. $v/(\mathrm{m\,s^{-1}}) = 1.54 \cdot (D/\mathrm{mm})^{0.61}$ | 0.95 | *[B]* |
| Graupel | – | 0–4.0 mm | 26. $v/(\mathrm{m\,s^{-1}}) = 1.25 \cdot (D/\mathrm{mm})^{0.94}$ | – | *[Le]* |

These are within or above *[VM]*'s confidence region. *[H]* reported lump graupel for temperatures below and above 0.5 °C, with

faster speeds for higher temperatures. These are below and above *[VM]*'s confidence region, respectively. Their relationship




for colder temperatures than 0.5 °C is closer to *[VM]*. The relationship by *[Le]* is close to *[VM]*, just above the confidence region, and the relationship from *[B]* is again at higher speeds, similar to the differences for plates and stellar.

In general, our $v$ vs. $D$ relationships agree reasonably well with the previous studies. The studies with the largest disparity compared to this study may, in large part, be explained by the different approach to classifying snow particle shapes.

## 4   Summary and conclusions

We have presented D-ICI measurements of natural snow, ice crystals and other hydrometeors, covering sizes from 0.06 to 3.2 mm. These data with dual images of every particle enable the retrieval of the particle shape, as well as size parameters from the top view and fall speed from the double-exposed side-view images.

The particles were sorted according to a classification scheme presented in Vázquez-Martín et al. (2020), which uses 15 different shape groups: needles, crossed needles, thick columns, capped columns, plates, stellar crystals, bullet rosettes, branches, spatial plates, spatial stellar, graupel, ice particles, irregulars, and spherical particles. In this study, we have analysed fall speed versus particle size ($v$ vs. $D_{\max}$) and fall speed versus cross-sectional area ($v$ vs. $A$) for each of the 15 shape groups. Fall speed dependence of particle orientation has also been studied as well as dependence of area ratio. Following is a summary of the conclusions drawn.

- Power-law functions represent the relationship between the cross-sectional area and the maximum dimension (Eq. (2)) very well for all shape groups (see Table 1). The exponent $b$ varies between about 1 and 2. Theoretically, the value approaches 1.0 for very elongated shapes that predominantly grow in only one of the two dimensions shown on the top-view images and 2.0 for spherical shapes. Indeed, data from the shape groups with very elongated shapes, *(1) Needles*, *(2) Crossed needles*, and *(3) Thick columns*, and the groups with round particles, *(12) Graupel* and *(15) Spherical*, have $b$ values close to these theoretical limits. For the other shape groups, $b$ varies between 1.4 and 1.8. Ultimately, as can be seen in Eq. (3), the smaller the value in $b$, the faster $A_r$ decreases as $D_{\max}$ increases (see Figure 4).

- Shape groups *(7) Bullet rosettes* and *(12) Graupel* have the fastest fall speeds with median speeds near 0.58 m s$^{-1}$ (see Sect. 3.2). The lowest median values of 0.34 m s$^{-1}$ or less are observed for shape groups *(1) Needles*, *(2) Crossed needles*, and *(3) Thick columns* and the median of all data is approximately 0.43 m s$^{-1}$, and most shape groups have their median within $\pm 0.08$ m s$^{-1}$ from this value.

- Overall, the fall speed data of individual particles show a broad spread of values as a function of $D_{\max}$ or $A$ so that no good correlation to the power-law fits given by Equations 4 and 5 exists. However, binning the data before applying the power law improves the correlations substantially. For all shape groups, the fit to the individual data and the fit to the data after binning, agree with each other within uncertainties. For about half of the shape groups, the correlation coefficients after binning the data is larger than 0.5, and the corresponding fits are considered representative. For the remaining groups, it is uncertain if it is possible to find sufficiently representative power-law fits. See Table 3 (for $v$ vs. $D_{\max}$) and Table 4 (for $v$ vs. $A$) for a full overview of these results.





- For the majority of shape groups, the $v$ vs. $A$ correlation is about equally good as $v$ vs. $D_{\max}$. This is expected due to the generally very good correlation between $A$ and $D_{\max}$.


- For a few shapes the $v$ vs. $D_{\max}$ and $v$ vs. $A$ correlations are different. For most of these shapes the $v$ vs. $A$ correlation is better than the $v$ vs. $D_{\max}$. The fall speed depends on mass and drag, and drag on the cross-sectional area so that one expects $A$ to be more significant for fall speed than $D_{\max}$.

- The drag force depends on cross-sectional area, but also on the particle Reynolds number, which in turn depends on a characteristic particle length. While for most shapes this characteristic length may be well approximated by $D_{\max}$, it

can be significantly different from $D_{\max}$ for a few shapes. For such shapes, one can expect low correlation for the $v$ vs. $D_{\max}$ relationship, and this is the case for shape groups *(1)–(3)*, for which $D_{\max}$ is equivalent to the needles' or columns' length, but the characteristic length is given by their width instead. These groups have low $v$ vs. $D_{\max}$ correlation but better $v$ vs. $A$ correlation.

- In this dataset, generally, only a few groups contained particles where we could distinguish clearly the orientation of

the falling particle, the planar and elongated shape groups. Only 135 particles have been found with close to exactly horizontal and vertical orientation. Of these, most are falling with a horizontal orientation, and we have found only 26 particles that are falling vertically orientated. These are falling slightly faster (the median is $0.08\ \mathrm{m\,s^{-1}}$ faster) than the horizontally orientated particles (see Sect. 3.3), however, the small sample size inhibited any further analysis of fall speed dependence on particle orientation.


- The shape groups *(1) Needles*, *(2) Crossed needles*, and *(3) Thick columns* show a distinct fall speed dependence on area ratio. By splitting particles of the same size or cross-sectional area into three categories of area ratio (low, medium, high), we found that those with larger area ratios have higher fall speeds. These relationships have a high correlation, and much higher than before splitting the data into different area ratio ranges (see Table 5). Only these three shape groups show this behaviour. Thus, if a similar area ratio dependence exists for other shapes, then it is less pronounced.


- Our $v$ vs. $D_{\max}$ relationships for some of the better-correlated shape groups, *(5) Plates*, *(6) Stellar*, and *(12) Graupel*, are compared with other fall speed relationships given by previous studies. Our results agree reasonably well with the studies that determined shape and fall speed for all particles or based on literature area–dimensional and mass–dimensional relationships for specific shapes. Of these studies, some of them are somewhat faster, and some are somewhat slower than our relationships for the corresponding shape group. Other studies differ significantly from our relationships. However,

in these studies, the shape groups were determined based on the identity of the most frequent particle shape within a time interval, i.e., other particle shapes undoubtedly reduced the precision of the dataset, and therefore may be the cause of the bias between this dataset and theirs (see Sect. 3.4).

These resulting parameterizations of the snow microphysical properties as a function of particle shape may be useful for improving our understanding of precipitation in cold climates in addition to helping improve the microphysical parameterizations

in the climate and forecast models.



*Data availability.* The presented data will be available at the Swedish National Data Service (DOI will be available).

*Author contributions.* Conceptualization, T.K., S.V.-M.; resources–instrumentation, T.K.; investigation, S.V.-M., T.K., S.E.; formal analysis, S.V.-M., T.K.; data curation, S.V.-M., T.K.; writing–original draft preparation, S.V.-M.; writing–review and editing, S.V.-M., T.K., S.E.; visualization, S.V.-M.; supervision, T.K., S.E.

*Competing interests.* The authors declare no conflict of interest.

*Acknowledgements.* We like to thank the Graduate School of Space Technology at the Luleå University of Technology for financial support and the Swedish Institute of Space Physics (IRF) at Kiruna for offering its facilities for our instrument.





## Appendix A: Fall speed relationships for the shape groups

Figures A1 and A2 represent $v$ vs. $D_{\max}$ and $v$ vs. $A$, respectively fitted to a power law, for all the 15 shape groups and, as in

Fig. 6 (Sect. 3.2), both methods ($M_{\mathrm{a}}$, $M_{\mathrm{b}}$) are shown for comparison.







**Figure A1.** Fall speed versus particle size ($v$ vs. $D_{\max}$) relationships given by Eq. (4) for all the shape groups are shown. Individual data (coloured symbols) and binned data (blue symbols with error bars) are displayed. Median values in the respective bins represent the binned data. The total length of the error bars represents the spread in fall speed data, which is given by the difference between the $16^{\mathrm{th}}$ and $84^{\mathrm{th}}$ percentiles. Fits that apply to individual data ($M_a$) and to binned data ($M_b$) are shown for comparison. The 68% prediction band and the 68% confidence region for both fits ($M_a$, $M_b$) are also shown. The same data are shown in Table 3.





**Figure A2.** Same as for Fig. A1, but fall speed versus cross-sectional area ($v$ vs. $A$) relationships given by Eq. (5) are shown here.



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
