# Peer review of "Shape dependence of snow crystal fall speed"

_Atmospheric Chemistry and Physics, 2020_

## Referee Comment (RC1) · Anonymous Referee #1 · 2 Dec 2020

As an overview, this paper presents what I think is an informative analysis of snow particle fall speed and its dependence on particle shape, maximum dimension and area ratio. The observations were collected using in situ imaging during winter seasons over a five-year period, giving fairly robust sampling for a number of particle shapes. Aside from the concerns I mention below, the results are well-supported by the analysis and interpretation. I have several more substantial concerns that I've described in my general comments below. These are followed by more specific comments. Overall, I think the work is valuable, with application to microphysical parameterization, process studies and remote sensing. I think the content is suitable for ACP and likely publishable after comments are addressed.

General comments

[Figure]

############

One of the most interesting aspects of this paper is the examination of whether v vs. D_max or v vs. A do a better job of predicting particle fall speeds, but this aspect of the work isn't highlighted in the abstract or introduction. I'd suggest adding this.

It is unclear to me how the method used to analyze images for fall speed in this work differs from the method presented in Kuhn and Vazquez-Martin (2020). The description in this paper (e. g., around lines 106 - 125) seems to suggest that an automated method is used when side-viewed particle exposures do not overlap, but a manual method is used when they do overlap. For example, at lines 109 - 110: "...whereas, in Fig. 1b, the particles are partly overlapping, which poses a limitation for an automated fall speed determination." But Kuhn and Vazquez-Martin (2020) say that fall speed determination is done using a manual method. It would be helpful to clarify (maybe succinctly describe the Kuhn and Vazquez-Martin (2020) method, then describe what is different in this work).

The paper lacks some essential detail about how the power law fits were performed, stating for example, simply that the parameters are "determined from linear fits to the data expessed as ...". I assume the data were log-transformed, then a linear least-squares fit was applied. How were uncertainties in the data treated? See my comment for lines 187-190, in particular.

Starting in the paragraph just before section 3.2.3 and within section 3.2.3 itself, there seems to be some confusion and lack of clarity about the meaning of $(R^2)_a$ and $(R^2)_b$ when compared to how they are shown in Tables 3 and 4. This affects the interpretation of the results. Perhaps consider reorganizing this section to discuss separately the effects on $R^2$ of binning and the effects on $R^2$ of using D_max or A as the independent variable for fall speed parameterization.

Specific comments

############

Lines 2-3: Note that the influence of shape depends strongly on radar wavelength, with precipitation radar reflectivities (at S- and C-band) being much less sensitive to shape than are cloud radars (at W- and Ka-band).

Line 6: It is unclear what "They" is referring to.

Line 7-8: This seeems to repeat the earlier statement at lines 2-3. Perhaps this should be removed?

*Lines 15-17: Not sure that these methodology details belong in the abstract. Maybe focus more on principal results.

Line 93: It's not clear to what number the 23% reduction applies. The previously-mentioned count was 10,000 particles, and a 23% reduction would leave 7,700 particles.

Lines 104-105: It may not be apparent to the general reader why these quantities are more relevant (and more relevant to what?) when observed from a vertical rather than horizontal viewing geometry.

Lines 106-110: Are multiple particles ever side-view imaged? I'm wondering how the automated fall speed method deals with this. Since that could be a source of error to fall speed determination, it would probably be relevant to discuss that briefly.

Lines 112-113: And for the automated fallspeed method, how does the method find these matching points? This question ties back to my initial comments.

Line 125: How much impact did this have on the previously-mentioned (line 93) sample size of 2,461 particles?

Line 145-147: It seems like "shape" is being used somewhat inconsistently here. The "shape groups" mentioned in line 145 refers to the 3-dimensional characteristics of the particle, I think. But when it is said here (line 147, "shape does not change with size"),

[Figure]

I think it's describing a 2-D property of the particle (the particle's projected area and area ratio). I suggest being more precise with the term - a particle shape can change with size while the area ratio remains constant.

Table 1: It would be useful to also know the RMS error of the fitted line versus the measured area ratios for each shape group. This would be useful, for example, for assessing uncertainties in fallspeed parameterizations that use area ratios. Could these be added to Table 1?

Line 152: This is more of a writing style comment, but I think it's usually not necessary to open with a statement of what the section contains. It's more engaging to proceed directly into the content (see for example, how you opened section 3.1).

Lines 152-156: These four lines mostly restate information already provided in the caption for Figure 5. Perhaps this could be made more succinct with something like "Analysis of the shape dependence of fall speed (Fig. 5) shows that shape groups (7) Bullet rosettes ...". Also, note that the sentence "Followed by shape groups (4) ..." is an incomplete sentence.

Figure 3 and 4 captions: Should be "length of the fit lines", I think.

Line 183: It would be useful to comment on the source for this randomness in determining the fall speed of individual particles, either here or (mabye better) in the discussion section. Is it due to errors in determining matching points in the particle images? Air currents within the device? Variations in particle orientation that cause the horizontally-projected area to vary?

Lines 187-190: Does this method of fitting use the scatter of fall speeds within each bin as an uncertainty for the bin fall speed (the median value)? It seems with such significant scatter in the individual data points, it would be important to de-emphasize bins with a lot of scatter when fitting. Also, if the bin widths become large (e. g., due to the requirement that all bins contain equal numbers of particles), it may be necessary

to also treat scatter in the particle sizes within each bin as an uncertainty on the median particle size for each bin. In cases with uncertainties in both the x- and y-values to be fitted, a method such as orthogonal distance regression is needed.

Lines 191-193: I'm not sure I agree completely with this assertion. Imagine an extreme case of fitting using M_b with only two bins. I wouldn't expect the results to match those from an M_a fit. Is there a reason that 10 bins were selected?

Line 201-208: This is saying that $(R^2)$_a is for v vs. D_max, and $(R^2)$_b is for v vs. A. Is this what is intended? That doesn't seem consistent with what is shown in Tables 3 and 4. Instead, Tables 3 and 4 show that $(R^2)$_a corresponds with M_a, and $(R^2)$_b corresponds to M_b. My understanding, then, is that you want $(R^2)$_b to be better than $(R^2)$_a in order to believe that M_b provides a vaild fit. Or do you want both correlations to be high? I think it is necessary to review the discussion in these lines to make sure it says what you intended. See my opening comments.

Line 211-212: Given the data in Tables 3 and 4, it's not clear to me how binning improved both $(R^2)$_a and $(R^2)$_b for capped columns. $(R^2)$_a doesn't represent binned fits. See my opening comments.

Lines 213-214: Similarly here, binning doesn't impact $(R^2)$_a, so I'm not sure what is meant by "improved their correlation coefficients $(R^2)$_a...". See my opening comments.

Line 223: Isn't cross-sectional area an aspect of particle size? Do you mean that fall speed depends more strongly on cross-sectional area than on maximum dimension?

Figure 7 caption: Again, this usage of $(R^2)$_a and $(R^2)$_b seems inconsistent with how they are represented in Tables 3 and 4.

Lines 242: Does the "(15) Spherical" shape group then represent solid ice spheres?

Lines 272-274: For me, the most interesting aspect of this is that the vertically-oriented particles do not fall *much* faster than the horizontally-oriented particles. Per, for example, Mitchell and Heymsfield (2005), fallspeed should depend on the ratio of mass to horizontally-projected area. It would be interesting to see the comparisons shown in Figure 11, but resolved by particle shape. Do vertically-oriented plates fall faster than horizontally-oriented plates of the same D_max? I'm not suggesting that this be included in this paper, of course, but perhaps for future work.

Line 285: I'm not sure what is meant by "overlapping somewhat in D_max and A".

Lines 293-319: This analysis is highly interesting, but it took several read-throughs to follow the development. Since mass, area and area ratio can all be expressed succinctly using power laws as functions of D_max, I wonder if it might be clearer to develop, for example, the resulting power law relationships for mass and area as functions of area ratio then show, by substituting in the appropriate exponents for elongated particles, that mass increases more rapidly than does A as A_r increases. Is there a reason this approach wasn't used?

Lines 308-311: If I look at prior studies like Mitchell (1996), it seems more common for other shapes to have mass power laws where the exponents are in the range of 1.7 to 2.4, rather than 3. Does this alter the interpretation presented here?

Figure 13 caption: Please also describe the purpose of the numbers labeling each line.

Table 6: For comparison purposes, I suggest that you consider converting all the relationships reported here to consistent units. I know it will affect only the leading coefficients, but it would make it easier to quantify the differences between the relationships.

---

## Referee Comment (RC2) · Anonymous Referee #2 · 8 Dec 2020

General Comments:

Studies that directly measure the fall speeds of naturally formed ice particles along with other properties that affect fall speeds are rare these days, and this paper is one of those rare papers. Obtaining ice particle imagery from two orthogonal viewing angles (top-down and horizontal) adds information not normally found in other studies, and allows for a more complete analysis of the fall speed physics. This adds new statistics to the orientation of ice crystals that have a preferred orientation (due to a much greater projected area associated with a given growth axis, such as found with hexagonal columns and plates). I am only aware of one other observational study that addresses this topic (Kajikawa, 1992, J. Meteor. Soc. Japan). The authors frame their results nicely in the context of other studies. Overall, the paper is well written and or-

ganized, and advances our knowledge of solid precipitation fall speeds. Some specific comments listed below should be addressed before the paper is published in ACP.

Specific Comments:

1. In the abstract, the points in the first "introduction" paragraph could be organized better and stated more succinctly, perhaps in about three sentences.

2. Lines 43 – 49: The recent work of Dunnavan (2020, JAS) should also be mentioned here as he shows how orientation and other factors account for much of the dispersion in ice fall speed for a given size.

3. Lines 50 – 52: Another reference that supports this is Mitchell et al. (2008, GRL), which shows how the cirrus ice fall speed affects cirrus cloud coverage and ice water path, and the resulting impact on radiation in the Community Atmosphere Model version 3 (CAM3).

4. Figure 3 and elsewhere in paper where shape category 13 is mentioned: Shape category 13 is ambiguous since it appears this can refer to any kind of ice particle and it also includes melting/sublimating ice particles of any shape apparently. Is this merely a "dust bin" category that includes all the other shapes that defy classification?

5. Figure 3 and elsewhere in paper where shape category 14 is mentioned: Irregulars as defined in Lawson et al. articles are high density, blocky-type particles that are subdivided into small and large irregulars, but other authors may define them differently. How are they defined here? If irregulars here are similar to irregulars in Lawson et al. papers, then grouping them with aggregates may be a mistake, since unrimed aggregates are NOT high density; rather they are notoriously low density. Please show a photo of irregulars so readers will know what they look like.

6. Lines 175 – 181: Regarding the large spread in fall speeds, the recent paper by Dunnavan (2020, J. Atmos. Sci.) shows snowflake (i.e., crystal aggregate) fall speeds are most sensitive to particle shape differences (largely accounted for in this study) with

secondary sensitivity to aggregate orientation. Please consider whether this Dunnavan study is relevant in explaining the large spread in fall speeds encountered here.

7. Figure 6: The legend contains a pink bar denoting the 68% confidence region, but a pink region is not evident in the plots. Perhaps it is sufficient to only show the 68% prediction bands?

8. Section 3.3.1 on Orientation: A study by Kajikawa is the only one I know of containing observational results on natural ice crystal orientation during free-fall; please include this reference (Kajikawa, 1992, J. Meteorol. Soc. Japan), comparing orientation results from this study with the Kajikawa results.

9. Section 3.4 (Comparison with previous fall speed relationships): Ice fall speeds depend on temperature T and pressure P; what were the values assumed here? Was this assumption applied universally to all of the studies in this intercomparison? If not, please ensure that all fall speeds in this intercomparison use the same T and P.

10. Figure 13, middle panel (stellar): Please mention that Curve 11 is based on the flow regime for larger ice particles, and should be closer to [VM] for the smaller D if the corresponding flow regime constants in [M] were used.

11. Lines 343-4: Sentence structure appears awkward; do you mean to say "Our shape group (5) contained only plates that were unrimed"?

Please also note the supplement to this comment:
https://acp.copernicus.org/preprints/acp-2020-1056/acp-2020-1056-RC2-supplement.pdf

———————————————

---

## Author Comment (AC1) · 15 Feb 2021

Dear referee #1,

We sincerely appreciate your constructive feedback and time spent to read and evaluate this work. Please look at the lines at the end of this text that contain our responses to your comments.

Best regards,

Sandra et al.
* * *
* * *
[Figure]
We sincerely appreciate the referee #1 for your constructive feedback and time spent to read and evaluate this work. Please see below our responses to your comments.

\*General comments: ——————————-

1) One of the most interesting aspects of this paper is the examination of whether v vs D_max or v vs A do a better job of predicting particle fall speeds, but this aspect of the work isn't highlighted in the abstract or introduction. I'd suggest adding this.

Thank you for pointing this out. We will add this finding to the Abstract and mention this aspect in the Introduction.

2) It is unclear to me how the method used to analyze images for fall speed in this work differs from the method presented in Kuhn and Vazquez-Martin (2020). The description in this paper (e. g., around lines 106 - 125) seems to suggest that an automated method is used when side-viewed particle exposures do not overlap, but a manual method is used when they do overlap. For example, at lines 109 - 110: "...whereas, in Fig. 1b, the particles are partly overlapping, which poses a limitation for an automated fall speed determination." But Kuhn and Vazquez-Martin (2020) say that fall speed determination is done using a manual method. It would be helpful to clarify (maybe succinctly describe the Kuhn and Vazquez-Martin (2020) method, then describe what is different in this work).

The method in this paper does not differ from Kuhn and Vázquez-Martín (2020). Thus, in both studies, the particle shape classification and the determination of fall speed are conducted manually.

We will clarify this in lines 106-111 by removing "which poses a limitation for an automated fall speed determination" and adding "In both cases, a manual procedure is carried out for the fall speed determination".

3) The paper lacks some essential detail about how the power law fits were performed, stating for example, simply that the parameters are "determined from linear fits to the data expressed as ...". I assume the data were log-transformed, then a linear least-squares fit was applied. How were uncertainties in the data treated?

You assumed correctly that we have used linear least-square fits. Then, lines 164-169 will be modified for clarity. Doing the fitting, we have not considered uncertainties or spread in the data. See also our responses to your comment 15).

4) Starting in the paragraph just before section 3.2.3 and within section 3.2.3 itself, there seems to be some confusion and lack of clarity about the meaning of $(\ddot{R}\ddot{E}2)\_a$ and $(\ddot{R}\ddot{E}2)\_b$ when compared to how they are shown in Tables 3 and 4. This affects the interpretation of the results. Perhaps consider reorganizing this section to discuss separately the effects on $\ddot{R}\ddot{E}2$ of binning and the effects on $\ddot{R}\ddot{E}2$ of using D_max or A as the independent variable for fall speed parameterization.

We have already organized the discussions so that effects on $R^2$ of binning (Sect. 3.2.2) and the effects on $R^2$ of using D_max or A as the independent variable (Sect. 3.2.3) are in separate sections. Our inconsistent choice of names for $R^2$ and error in Tables 3 and 4 are the source of your confusion, we apologize for that. We will rename R_a to R_D since it refers to the v vs D_max relationships and R_b to R_A since it refers to v vs A. In Table 3 we will use the name R_D and in Table 4 R_A. To avoid ambiguity, we will also modify the text when we first mention R_D and R_A.

*Specific comments: ——————————-

1) Lines 2-3: Note that the influence of shape depends strongly on radar wavelength, with precipitation radar reflectivities (at S- and C-band) being much less sensitive to

shape than are cloud radars (at W- and Ka-band).

The influence, as you have pointed out, depends on radar wavelength and particle size. We will avoid mentioning radar as these details are not needed for our introduction. Accordingly, we will remove that sentence. In this way we will also address your comment 3) and a comment by referee 2 (general comment 1). In addition, we will edit Lines 26-32 in Introduction in order to make the text more concise and avoid ambiguities talking about remote sensing applications and precipitation retrievals that are not addressed in this work.

2) Line 6: It is unclear what "They" is referring to.

Line 6 will be corrected as follows:

They are Fall speed is also required for snowfall predictions . . .

3) Line 7-8: This seems to repeat the earlier statement at lines 2-3. Perhaps this should be removed?

The sentence in line 2-3 will be removed (see specific comment 1 above).

4) Lines 15-17: Not sure that these methodology details belong in the abstract. Maybe focus more on principal results.

We will remove the methodology details and leave what is important: Relationships between particle size, cross-sectional area, and fall speed are studied for different shapes.

5) Line 93: It's not clear to what number the 23% reduction applies. The previously-mentioned count was 10,000 particles, and a 23% reduction would leave 7,700 particles.

Not all the mentioned >10,000 particle images have been analysed yet. As stated, we have selected part of these data. After excluding FOV and tumbling problems, we are left with about 3,200 images. Of these, 23% were measured at wind speeds >3m/s.

This will be clarified in the text.

6) Lines 104-105: It may not be apparent to the general reader why these quantities are more relevant (and more relevant to what?) when observed from a vertical rather than horizontal viewing geometry.

We will state why they are more relevant for drag, and thus fall speed, and for comparison with remote sensing data.

7) Lines 106-110: Are multiple particles ever side-view imaged? I'm wondering how the automated fall speed method deals with this. Since that could be a source of error to fall speed determination, it would probably be relevant to discuss that briefly.

Our fall speed measurements are determined manually not by an automated method. See also 2) in general comments.

8) Lines 112-113: And for the automated fallspeed method, how does the method find these matching points? This question ties back to my initial comments.

The matching points are found manually. Again, see 2) in general comments.

9) Line 125: How much impact did this have on the previously-mentioned (line 93) sample size of 2,461 particles?

Due to our manual analysis, this did not have any impact. See 2) in general comments.

10) Line 145-147: It seems like "shape" is being used somewhat inconsistently here. The "shape groups" mentioned in line 145 refers to the 3-dimensional characteristics of the particle, I think. But when it is said here (line 147, "shape does not change with size"), I think it's describing a 2-D property of the particle (the particle's projected area and area ratio). I suggest being more precise with the term - a particle shape can change with size while the area ratio remains constant.

You are right, the conclusion here was about "shape" in terms of area ratio. Lines 146-147 will be modified accordingly: Thus, apart from (15) Spherical, (12) Graupel is the

only shape group where Ar remains constant with increasing Dmax.

11) Table 1: It would be useful to also know the RMS error of the fitted line versus the measured area ratios for each shape group. This would be useful, for example, for assessing uncertainties in fallspeed parameterizations that use area ratios. Could these be added to Table 1?

Yes, we agree that including RMS errors is useful to show the uncertainty in each group. Then, we will add RMS errors for both calculated Ar and fitted A in that table as well as for v in Tables 3 and 4.

12) Line 152: This is more of a writing style comment, but I think it's usually not necessary to open with a statement of what the section contains. It's more engaging to proceed directly into the content (see for example, how you opened section 3.1).

Thank you, we agree. The opening statement in line 152 will be removed.

13) Lines 152-156: These four lines mostly restate information already provided in the caption for Figure 5. Perhaps this could be made more succinct with something like "Analysis of the shape dependence of fall speed (Fig. 5) shows that shape groups (7) Bullet rosettes ...". Also, note that the sentence "Followed by shape groups (4) ..." is an incomplete sentence.

Thank you, we agree and will change this part accordingly.

14) Figure 3 and 4 captions: Should be "length of the fit lines", I think.

Thank you, we agree. The captions have been corrected.

15) Line 183: It would be useful to comment on the source for this randomness in determining the fall speed of individual particles, either here or (maybe better) in the discussion section. Is it due to errors in determining matching points in the particle images? Air currents within the device? Variations in particle orientation that cause the horizontally-projected area to vary?
We have discussed the uncertainties of the measurement methods in our previous work (Kuhn & Vazquez Martín, 2020). While these certainly contribute to the spread in data, most of that is likely inherent to the type of data, i.e., fall speed that depends on particle drag, which in turn depends on shape and orientation. We will extend the sentence in Lines 189-190 to include a short discussion about this.

16) Lines 187-190: Does this method of fitting use the scatter of fall speeds within each bin as an uncertainty for the bin fall speed (the median value)? It seems with such significant scatter in the individual data points, it would be important to de-emphasize bins with a lot of scatter when fitting. Also, if the bin widths become large (e. g., due to the requirement that all bins contain equal numbers of particles), it may be necessary to also treat scatter in the particle sizes within each bin as an uncertainty on the median particle size for each bin. In cases with uncertainties in both the x- and y-values to be fitted, a method such as orthogonal distance regression is needed.

We have not used the scatter within each bin as a measure of uncertainty in the fitting method. Instead, we have selected our simpler fitting method. When we started fitting after binning the data, we first tested equally-spaced $D_{max}$ or $A$ bins, however, frequently observed issues with the first and/or last bin when only one or two particles were included in the bin. Changing bin sizes would then affect the fitted relationships. We could resolve these issues by requiring an equal number of particles in each bin. The number of bins, ten, was a compromise we have selected between many bins but too few particles per bin and many particles in each bin but too few bins for a good fit.

17) Lines 191-193: I'm not sure I agree completely with this assertion. Imagine an extreme case of fitting using $M_b$ with only two bins. I wouldn't expect the results to match those from a $M_a$ fit. Is there a reason that 10 bins were selected?

As explained in 16), the number is a compromise to have enough bins to build a regression, but few enough so that there are enough particles in each bin to get a representative median to decrease the spread in data within each bin. With this compromise,

we expect Mb and Ma to agree reasonably well, and in the following, we discuss for which shape groups this is the case and for which not. We will change the text to make this clearer.

18) Line 201-208: This is saying that $(R^2)\_a$ is for v vs D_max, and $(R^2)\_b$ is for v vs A. Is this what is intended? That doesn't seem consistent with what is shown in Tables 3 and 4. Instead, Tables 3 and 4 show that $(R^2)\_a$ corresponds with M_a, and $(R^2)\_b$ corresponds to M_b. My understanding, then, is that you want $(R^2)\_b$ to be better than $(R^2)\_a$ in order to believe that M_b provides a valid fit. Or do you want both correlations to be high? I think it is necessary to review the discussion in these lines to make sure it says what you intended. See my opening comments.

Yes, that was what we intended. And you are right it is inconsistent with Tab 3 and 4. See our response to 4) in general comments. In addition to the changes mentioned there, we will rephrase a few sentences here to make our intentions clearer, i.e. that we are looking at $R^2$ for Method b to check for which of the shape groups the fitted relationships for v vs Dmax and v vs A are reliable.

19) Line 211-212: Given the data in Tables 3 and 4, it's not clear to me how binning improved both $(R^2)\_a$ and $(R^2)\_b$ for capped columns. $(R^2)\_a$ doesn't represent binned fits. See my opening comments.

The confusion should be resolved by our response to 4) in general comments. $R^2$ for v vs Dmax improved from 0.12 (Ma) to 0.42 (Mb). For v vs A the improvement is from 0.10 to 0.49.

20) Lines 213-214: Similarly here, binning doesn't impact $(R^2)\_a$, so I'm not sure what is meant by "improved their correlation coefficients $(R^2)\_a$...". See my opening comments.

See again our response to 4) in general comments.

21) Line 223: Isn't cross-sectional area an aspect of particle size? Do you mean that fall

speed depends more strongly on cross-sectional area than on maximum dimension? Figure 7 caption: Again, this usage of (RËĘ2)_a and (RËĘ2)_b seems inconsistent with how they are represented in Tables 3 and 4.

We use particle size and maximum dimension interchangeably. For clarity, we will change "particle size" here to "maximum dimension" or change the text to make the interchangeability clearer. Regarding Fig. 7 caption, please see the answer to 4) in general comments.

22) Lines 242: Does the "(15) Spherical" shape group then represent solid ice spheres?

Shape group (15) Spherical includes frozen small raindrops and liquid raindrops. These particles have spherical shape and morphology but are solid or liquid particles as shown in Fig. 1 attached as a pdf (see more details in Vázquez-Martín et al., 2020 "Shape Dependence of Falling Snow Crystals' Microphysical Properties Using an Updated Shape Classification"). H1c are ice particles and H4a are liquid droplets. If the particle is a perfect sphere, it is assumed to be liquid and thereby considered a raindrop, whereas if the particle has a popping bubble on the edge, it is considered as an ice particle.

23) Lines 272-274: For me, the most interesting aspect of this is that the vertically-oriented particles do not fall *much* faster than the horizontally-oriented particles. Per, for example, Mitchell and Heymsfield (2005), fallspeed should depend on the ratio of mass to horizontally-projected area. It would be interesting to see the comparisons shown in Figure 11, but resolved by particle shape. Do vertically-oriented plates fall faster than horizontally-oriented plates of the same D_max? I'm not suggesting that this be included in this paper, of course, but perhaps for future work.

Thanks for your comment and suggestion, very interesting. We had investigated what you mention for different shapes individually, i.e., for shape groups (1) Needles, (5) Plates and (6) Stellar. However, probably due to the small sample size, the results were not conclusive showing larger uncertainties than differences between the two

falling orientations. We will certainly look at these questions again in future studies when more data will be available.

24) Line 285: I'm not sure what is meant by "overlapping somewhat in D_max and A".

We will rephrase to make this clearer:

As can be seen in Fig. 12, these relationships are spread out in a way, so that for a given...

25) Lines 293-319: This analysis is highly interesting, but it took several read-throughs to follow the development. Since mass, area and area ratio can all be expressed succinctly using power laws as functions of D_max, I wonder if it might be clearer to develop, for example, the resulting power law relationships for mass and area as functions of area ratio then show, by substituting in the appropriate exponents for elongated particles, that mass increases more rapidly than does A as A_r increases. Is there a reason this approach wasn't used?

The purpose of this section is to investigate the Ar dependence observed in Fig 12. For that, we are considering the special case of a subset of the data with constant Dmax. Because of that, we cannot use normal power laws as a function of Dmax to do this investigation. If one would use them, then the particle mass was constant. We will modify the text slightly to improve clarity. See also our response to your comment 26), where we correct an error.

26) Lines 308-311: If I look at prior studies like Mitchell (1996), it seems more common for other shapes to have mass power laws where the exponents are in the range of 1.7 to 2.4, rather than 3. Does this alter the interpretation presented here?

We examine here the special case of constant Ar. For this A is proportional to D^2 as stated. However, it was wrong to state that m would then be proportional to D^3. There could, of course, be shapes such as simple plates where Ar is constant and where thickness (not seen on image) does not increase proportionally to D. In an extreme

case, m would then be proportional to $D^2$. So, in general, in a dataset with constant Ar, m would be proportional to anything between $D^2$ to $D^3$. We will change the text to correct this. It does not alter the interpretation presented.

27) Figure 13 caption: Please also describe the purpose of the numbers labeling each line.

The purpose of labelling each line was to easily connect the fit lines to the power laws displayed in Table 6. We will mention this in both captions of Table 6 and Fig. 13.

28) Table 6: For comparison purposes, I suggest that you consider converting all the relationships reported here to consistent units. I know it will affect only the leading coefficients, but it would make it easier to quantify the differences between the relationships.

The reason why we have shown the relationships in different units was to be consistent with the power laws used from the literature. However, we agree that for comparison it would be more convenient if all used the same reference units. Therefore, we will be adding such information in the text, for instance, by adding "The power laws from the literature have been converted to the same units." Consequently, we will change the relationships shown in Table 6.

\*We made some text edits: ——————————————

1) Line 102 has been corrected as it follows:

"cross-sectional are" has been replaced by "cross-sectional area"

2) Replace versus 'vs.' by 'vs'.

——————————————————

[Figure]

**Fig. 1.** Specific comment, #22)

---

## Author Comment (AC2) · 15 Feb 2021

Dear referee #2,

We sincerely appreciate your constructive feedback and time spent to read and evaluate this work. Please look at the lines at the end of this text that contain our responses to your comments.

Best regards,

Sandra et al.
* * *
* * *
[Figure]
We sincerely appreciate the referee #2 for your constructive feedback and time spent to read and evaluate this work. Please see below our response to your comments.

*General comments: ————————————-

1) In the abstract, the points in the first "introduction" paragraph could be organized better and stated more succinctly, perhaps in about three sentences.

Thank you for pointing this out. We will modify the first "Introduction" paragraph in the Abstract in order to make it more concise and also better organized (see also the response to specific comment 1 of referee 1).

2) Lines 43 – 49: The recent work of Dunnavan (2020, JAS) should also be mentioned here as he shows how orientation and other factors account for much of the dispersion in ice fall speed for a given size.

Thanks for your suggestion. We will include that reference in the previous paragraph talking about the influence of the shape on the spread in fall speed, which in their study was more important than orientation.

3) Lines 50 – 52: Another reference that supports this is Mitchell et al. (2008, GRL), which shows how the cirrus ice fall speed affects cirrus cloud coverage and ice water path, and the resulting impact on radiation in the Community Atmosphere Model version 3 (CAM3).

Thank you for your suggestion. We will add Mitchell et al. 2008.

4) Figure 3 and elsewhere in paper where shape category 13 is mentioned: Shape

category 13 is ambiguous since it appears this can refer to any kind of ice particle and it also includes melting/sublimating ice particles of any shape apparently. Is this merely a "dust bin" category that includes all the other shapes that defy classification?

Our "dust bin" is the shape group (14). Group (13) is, as it says, melting and sublimating (mostly small) particles as shown in Fig. 1 attached as a pdf (see more details in Vázquez-Martín et al., 2020 "Shape Dependence of Falling Snow Crystals' Microphysical Properties Using an Updated Shape Classification"). However, we think you are right in that shape group (14) should have better been divided into "Irregulars" (i.e. non aggregates) and "Aggregates", that's something we did not consider when first doing the shape classification.

5) Figure 3 and elsewhere in paper where shape category 14 is mentioned: Irregulars as defined in Lawson et al. articles are high density, blocky-type particles that are subdivided into small and large irregulars, but other authors may define them differently. How are they defined here? If irregulars here are similar to irregulars in Lawson et al. papers, then grouping them with aggregates may be a mistake, since unrimed aggregates are NOT high density; rather they are notoriously low density. Please show a photo of irregulars so readers will know what they look like.

Same as in 4), we think you are right in that shape group (14) should have better been divided into "Irregulars" (i.e. non aggregates) and "Aggregates", that's something we did not consider when first doing the shape classification. We have added pictures here (see Fig. 2 attached as a pdf) and referred readers in the MS to our previous work on shape classification, which contains many example images (Vázquez-Martín et al, 2020). In our dataset, we do not have many aggregates yet. However, in future, we will extend our dataset and then likely revise the shape grouping including a separate group for aggregates.

6) Lines 175 – 181: Regarding the large spread in fall speeds, the recent paper by Dunnavan (2020, J. Atmos. Sci.) shows snowflake (i.e., crystal aggregate) fall speeds

are most sensitive to particle shape differences (largely accounted for in this study) with secondary sensitivity to aggregate orientation. Please consider whether this Dunnavan study is relevant in explaining the large spread in fall speeds encountered here.

We will mention the spread in fall speed caused by shape variations (Dunnavan 2020) in the Introduction in response to your general comment 2). Responding to specific comment 15) of Referee 1, we will also extend the discussion on the reasons for the fall-speed spread in the next section on the binning method.

7) Figure 6: The legend contains a pink bar denoting the 68% confidence region, but a pink region is not evident in the plots. Perhaps it is sufficient to only show the 68% prediction bands?

This is indeed difficult to see. The confidence regions for Ma and Mb are shown as half-transparent pink and light blue, respectively. Here, they are almost entirely overlapping, making it difficult to make out where the pink region is. We will try to improve this, perhaps changing the v range to 0-1.0 m/s and removing the Mb prediction band will help. Otherwise, we will also consider your suggestion.

8) Section 3.3.1 on Orientation: A study by Kajikawa is the only one I know of containing observational results on natural ice crystal orientation during free-fall; please include this reference (Kajikawa, 1992, J. Meteorol. Soc. Japan), comparing orientation results from this study with the Kajikawa results.

Thank you, this is an interesting study. Kajikawa 1992 looked at falling motion with stable and unstable patterns, rather than at orientation directly as we did and found only small variations (few %) in the vertical speed with respect to the mean for individual trajectories. The smaller horizontal speeds showed more variations that Kajikawa linked to growth by aggregation.

9) Section 3.4 (Comparison with previous fall speed relationships): Ice fall speeds depend on temperature T and pressure P; what were the values assumed here? Was

this assumption applied universally to all of the studies in this intercomparison? If not, please ensure that all fall speeds in this intercomparison use the same T and P.

Thank you for raising this issue. We are aware of this fact and that different studies are based on measurements at different atmospheric conditions and may be adjusted to some normal conditions or not. Our measurement site is at an altitude of around 400 m, which corresponds to about a pressure difference of 50 hPa with respect to sea level. A pressure difference of 100 hPa (about 1000 m altitude difference) would result in about 5% difference in fall speed (see for example Khvorostyanov and Curry 2005, JAS 62, 4343-4357). A temperature difference of 10 K would cause a 2% speed difference. As these differences are relatively small compared to the typical speed differences between different studies we decided, for simplicity, to neglect these effects without any impact on the comparison. We will mention the altitude of our site and make a statement about the neglected P and T dependence of fall speed in the MS.

10) Figure 13, middle panel (stellar): Please mention that Curve 11 is based on the flow regime for larger ice particles, and should be closer to [VM] for the smaller D if the corresponding flow regime constants in [M] were used.

Thank you for pointing this out. We will mention that in the MS.

11) Lines 343-4: Sentence structure appears awkward; do you mean to say "Our shape group (5) contained only plates that were unrimed"?

The sentence will be modified accordingly and convey a little more detail.
* * *
[Figure]

| I1 | H1a | H1b | H2a | H3a | 1.0 |

**Fig. 1.** General comment, #4)

[Figure]

**Fig. 2.** General comment, #5)

---

## Author Response (AR2)

* * *
We sincerely appreciate the referee #1 for your constructive feedback and time spent to read and evaluate this work. Please see below our responses to your comments.

Changes done in the MS as a consequence of our responses are indicated; all line numbers refer to the MS as originally submitted.

- **General comments:**

**1) One of the most interesting aspects of this paper is the examination of whether v vs D_max or v vs A do a better job of predicting particle fall speeds, but this aspect of the work isn't highlighted in the abstract or introduction. I'd suggest adding this.**

Thank you for pointing this out. We will add this finding to the Abstract and mention this aspect in the Introduction.

MS changed in the Abstract as indicated.

L. 19, before sentence "The effects of orientation…" ADDED a new sentence:
"For most of these shapes the fall speed is better correlated with cross-sectional area than with particle size."

**2) It is unclear to me how the method used to analyze images for fall speed in this work differs from the method presented in Kuhn and Vazquez-Martin (2020). The description in this paper (e. g., around lines 106 - 125) seems to suggest that an automated method is used when side-viewed particle exposures do not overlap, but a manual method is used when they do overlap. For example, at lines 109 - 110: "...whereas, in Fig. 1b, the particles are partly overlapping, which poses a limitation for an automated fall speed determination." But Kuhn and Vazquez-Martin (2020) say that fall speed determination is done using a manual method. It would be helpful to clarify (maybe succinctly describe the Kuhn and Vazquez-Martin (2020) method, then describe what is different in this work).**

The method in this paper does not differ from Kuhn and Vázquez-Martín (2020). Thus, in both studies the particle shape classification and the determination of fall speed are conducted manually.

We will clarify this in lines 106-111 by removing "which poses a limitation for an automated fall speed determination" and adding "In both cases, a manual procedure is carried out for the fall speed determination".

L. 110-111 "… are partly overlapping, which poses a limitation for an automated fall speed determination. The following describes the manual procedure to, nonetheless, include such particles in the analysis."

CHANGED TO:

"… are partly overlapping. In both cases, a manual procedure is carried out for the fall speed determination, which is described in the following."

**3) The paper lacks some essential detail about how the power law fits were performed, stating for example, simply that the parameters are "determined from linear fits to the data expressed as ...". I assume the data were log-transformed, then a linear least-squares fit was applied. How were uncertainties in the data treated?**

You assumed correctly that we have used linear least-square fits. Then, lines 164-169 will be modified for clarity. Doing the fitting, we have not considered uncertainties or spread in the data. See also our responses to your comment 15).

MS changed as indicated.

L. 166 after Eq. (4) "which yields straight lines on a logarithmic plot, and, from these lines, derive the parameters $a_D$ and $b_D$."

CHANGED TO:

"The parameters $a_D$ and $b_D$ are determined from linear fits to the data expressed as logarithm of v vs logarithm of Dmax."

**4) Starting in the paragraph just before section 3.2.3 and within section 3.2.3 itself, there seems to be some confusion and lack of clarity about the meaning of $(R^2)\_a$ and $(R^2)\_b$ when compared to how they are shown in Tables 3 and 4. This affects the interpretation of the results. Perhaps consider reorganizing this section to discuss separately the effects on $R^2$ of binning and the effects on $R^2$ of using D_max or A as the independent variable for fall speed parameterization.**

We have already organized the discussions so that effects on $R^2$ of binning (Sect. 3.2.2) and the effects on $R^2$ of using D_max or A as the independent variable (Sect. 3.2.3) are in separate sections.
Our inconsistent choice of names for $R^2$ and error in Tables 3 and 4 are the source of your confusion, we apologize for that.
We will rename R_a to R_D since it refers to the v vs D_max relationships and R_b to R_A since it refers to v vs A. In Table 3 we will use the name R_D and in Table 4 R_A.
To avoid ambiguity, we will also modify the text when we first mention R_D and R_A.

MS changed as indicated.

In Tab 3 "R_a" and "R_b" CHANGED TO: "R_D".

In Tab 4 "R_a" and "R_b" CHANGED TO: "R_A".

Anywhere else, "R_a" CHANGED TO: "R_D", and "R_b" CHANGED TO: "R_A".

L. 196-197 "After binning, the correlation coefficients are much higher for both the v vs. Dmax and the v vs. A relationships with R2 $\simeq$ 0.89 (Table 3) and R2 $\simeq$ 0.87 (Table 4), respectively."

CHANGED TO:

"After binning, the correlation coefficients, which for clarity are denoted $R_D^2$ and $R_A^2$ for the fits to v vs $D_{max}$ and v vs A relationships, respectively, are much higher with $R_D^2 \simeq$ 0.88 (Table 3) and $R_A^2 \simeq$ 0.88 (Table 4)."

L. 201-202 "For clarity, $R^2$ for the two types of comparisons are denoted differently so that $R_a^2$ and $R_b^2$ are the individual correlation scores for the fits to v vs. $D_{max}$ and v vs. A relationships, respectively."

We DELETED this sentence.

■ **Specific comments:**

**1) Lines 2-3: Note that the influence of shape depends strongly on radar wavelength, with precipitation radar reflectivities (at S- and C-band) being much less sensitive to shape than are cloud radars (at W- and Ka-band).**

The influence, as you have pointed out, depends on radar wavelength and particle size. We will avoid mentioning radar as these details are not needed for our introduction. Accordingly, we will remove that sentence. In this way we will also address your comment 3) and a comment by referee 2 (general comment 1).
In addition we will edit Lines 26-32 in Introduction in order to make the text more concise and avoid ambiguities talking about remote sensing applications and precipitation retrievals that are not addressed in this work.

MS changed as indicated.

L. 3-4 "In particular, the shape is an important parameter as it strongly influences the scattering properties of these ice particles, and thus their response to remote sensing techniques such as radar measurements."

DELETED this sentence.

L. 27-31 "Also, the accuracy of many different remote sensing applications, such as scattering properties, cloud and precipitation retrievals from satellite passive and active microwave measurements (e.g., Bauer et al., 1999; Olson et al., 2001), and snowfall estimates based on ground- and space-based

radar (Cooper et al., 2017), is highly dependent on the assumptions made on the microphysical properties of snow particles."

CHANGED TO:

"Also, the accuracy of many different remote sensing applications, such as cloud and precipitation retrievals from satellite passive and active microwave measurements (Posselt et al., 2008; Zhang et al., 2009; Cooper and Garrett, 2010, and others) is highly dependent on the assumptions made on the microphysical properties of snow particles."

**2)  Line 6: It is unclear what "They" is referring to.**

Line 6 will be corrected.

MS changed as indicated.

L. 6 "They are also required for snowfall predictions …"

CHANGED TO:

"Fall speed is also required for snowfall predictions …"

**3)  Line 7-8: This seems to repeat the earlier statement at lines 2-3. Perhaps this should be removed?**

The sentence in line 2-3 will be removed (see specific comment 1 above).

**4) Lines 15-17: Not sure that these methodology details belong in the abstract. Maybe focus more on principal results.**

We will remove the methodology details and leave what is important:
Relationships between particle size, cross-sectional area, and fall speed are studied for different shapes.

MS changed as indicated.

L. 15-17 "The particles are shape-classified into 15 different shape groups depending on their shape and morphology. For these 15 shape groups relationships are studied, firstly, between size and cross-sectional area, then between fall speed and size or cross- sectional area."

CHANGED TO:

"Relationships between particle size, cross-sectional area, and fall speed are studied for different shapes."

**5) Line 93: It's not clear to what number the 23% reduction applies. The previously-mentioned count was 10,000 particles, and a 23% reduction would leave 7,700 particles.**

Not all the mentioned >10,000 particle images have been analysed yet. As stated, we have selected part of these data. After excluding FOV and tumbling problems, we are left with about 3,200 images. Of these, 23% were measured at wind speeds >3m/s. This will be clarified in the text.

MS changed as indicated.

L. 92-93 "The site in Kiruna does not often experience high wind speeds; hence the dataset is only reduced by 23% to 2,461 particles from 48 days."

CHANGED TO:

"After excluding FOV and tumbling problems, about 3,200 particles remained. Of these, 23% were measured at wind speeds higher than 3 ms−1, leaving a total of 2,461 particles to form our dataset."

**6) Lines 104-105: It may not be apparent to the general reader why these quantities are more relevant (and more relevant to what?) when observed from a vertical rather than horizontal viewing geometry.**

We will state why they are more relevant for drag, and thus fall speed, and for comparison with remote sensing data.

MS changed as indicated.

L. 104-105: "These quantities are particularly relevant when provided from this vertical viewing geometry corresponding to the falling motion, rather than a horizontal geometry, which is standard in many instruments."

CHANGED TO:

"The aerodynamic drag, which chiefly governs fall speed, more closely depends on the quantities $D_{max}$, A, and $A_r$ retrieved using images that view the particles from above (in the falling direction) as done by D-ICI, rather than from a horizontal viewing direction as done by other instruments. Furthermore, this view is more suitable to enable comparison with remote sensing measurements that often also have a vertical viewing geometry."

**7) Lines 106-110: Are multiple particles ever side-view imaged? I'm wondering how the automated fall speed method deals with this. Since that could be a source of error to fall speed determination, it would probably be relevant to discuss that briefly.**

Our fall speed measurements are determined manually not by an automated method. See also 2) in general comments.

**8) Lines 112-113: And for the automated fallspeed method, how does the method find these matching points? This question ties back to my initial comments.**

The matching points are found manually. Again, see 2) in general comments.

**9) Line 125: How much impact did this have on the previously-mentioned (line 93) sample size of 2,461 particles?**

Due to our manual analysis, this did not have any impact. See 2) in general comments.

**10) Line 145-147: It seems like "shape" is being used somewhat inconsistently here. The "shape groups" mentioned in line 145 refers to the 3-dimensional characteristics of the particle, I think. But when it is said here (line 147, "shape does not change with size"), I think it's describing a 2-D property of the particle (the particle's projected area and area ratio). I suggest being more precise with the term - a particle shape can change with size while the area ratio remains constant.**

You are right, the conclusion here was about "shape" in terms of area ratio. Lines 146-147 will be modified accordingly:
Thus, apart from *(15) Spherical*, *(12) Graupel* is the only shape group where Ar remains constant with increasing Dmax.

MS changed as indicated.

L. 146-148: "Thus, apart from *(15) Spherical*, *(12) Graupel* is the only shape, for which shape does not change with size, i.e., $A_r$ does not decrease with increasing $D_{max}$ but remains constant."

CHANGED TO:

"Thus, apart from *(15) Spherical*, *(12) Graupel* is the only shape group where $A_r$ remains constant with increasing $D_{max}$."

**11) Table 1: It would be useful to also know the RMS error of the fitted line versus the measured area ratios for each shape group. This would be useful, for example, for assessing uncertainties in fallspeed parameterizations that use area ratios. Could these be added to Table 1?**

Yes, we agree that including RMS errors is useful to show the uncertainty in each group. Then, we will add RMS errors for both calculated Ar and fitted A in that table as well as for v in Tables 3 and 4.

MS changed as indicated.

Changes in Table 1:
Column with title "RMSE" added as last column of Table 1, besides column "$R^2$".
We added the following text to the end of the caption:
"The root-mean-square error (RMSE) values of base-10 logarithms of measured A vs predicted A are also shown to indicate the uncertainty of these power laws. Note that RMSE values of logarithms of Ar as determined from measurements using Eq. (1) vs predicted values using Eq. (3) are the same."

Changes in Table 3:

Column with title "RMSE" added as last column of Table 3, besides column "$R_D^2$".
We added the following text to the end of the caption:
"The root-mean-square error (RMSE) values of base-10 logarithms of measured v vs predicted v are also shown to indicate the uncertainty of these power laws."

Changes in Table 4:
Column with title "RMSE" added as last column of Table 4, besides column "$R_A^2$".
We added the following text to the end of the caption:
"The root-mean-square error (RMSE) values of base-10 logarithms of measured v vs predicted v are also shown to indicate the uncertainty of these power laws."

**12) Line 152: This is more of a writing style comment, but I think it's usually not necessary to open with a statement of what the section contains. It's more engaging to proceed directly into the content (see for example, how you opened section 3.1).**

Thank you, we agree. The opening statement in line 152 will be removed.

MS changed as indicated.

L. 152 "This section provides an analysis of how the particle size, cross-sectional area, and shape influence its fall speed."
Was DELETED.

**13) Lines 152-156: These four lines mostly restate information already provided in the caption for Figure 5. Perhaps this could be made more succinct with something like "Analysis of the shape dependence of fall speed (Fig. 5) shows that shape groups (7) Bullet rosettes ...". Also, note that the sentence "Followed by shape groups (4) ..." is an incomplete sentence.**

Thank you, we agree and will change this part accordingly.

MS changed as indicated. In addition, some information removed from the text has been moved to the caption of Fig. 5.

L. 152-157 "Figure 5 shows the distribution of fall speed data for each shape group and all data (regardless of shape). A vertical bar represents the distribution of fall speed data for each shape. The bounds of each bar correspond to the 16th and 84th percentiles of the
155 distribution. The point on the bar shows the location of the median value. These bounds would correspond to $\pm 1\sigma$ (standard deviation) if the distribution were normal. Table 2 contains a list of these percentiles and medians. We note that shape groups (7) Bullet rosettes and (12) Graupel have the fastest fall speeds with a median value of v $\simeq$ 0.6 m s$-1$ . Followed by shape groups (4)…"

CHANGED TO:

"Analysis of the shape dependence of fall speed (see Figure 5) shows that shape groups (7) Bullet rosettes and (12) Graupel have the fastest fall speeds with a median value of v $\simeq$ 0.6 m s$-1$ , followed by shape groups (4)…"

**14) Figure 3 and 4 captions: Should be "length of the fit lines", I think.**

Thank you, we agree. The captions have been corrected.

MS changed as indicated.

In both captions of Figure 3 and 4, the text
"The length the fit lines are defined…"
Was CHANGED TO:
"The length of the fit lines are defined…"

**15) Line 183: It would be useful to comment on the source for this randomness in determining the fall speed of individual particles, either here or (maybe better) in the discussion section. Is it due to errors in determining matching points in the particle images? Air currents within the device? Variations in particle orientation that cause the horizontally-projected area to vary?**

We have discussed the uncertainties of the measurement methods in our previous work (Kuhn & Vazquez Martín, 2020). While these certainly contribute to the spread in data, most of that is likely inherent to the type of data, i.e., fall speed that depends on particle drag, which in turn depends on shape and orientation.
We will extend the sentence in Lines 189-190 to include a short discussion about this.

MS changed as indicated.

L. 189-190 "Previous studies have also used a similar method of binning fall speed before data fitting (e.g., Barthazy and Schefold, 2006; Zawadzki et al., 2010)."

CHANGED TO:

"The apparent randomness in fall speed, manifested as the wide spread in data, may have several reasons. While instrumental uncertainties and errors introduced by the manual analysis (see Kuhn and Vázquez-Martín, 2020) contribute to the variability, much of the observed randomness is likely inherent to the data. For example, Dunnavan (2021) showed that aggregate snowflakes' fall speed is very sensitive to shape. Other studies have also reported a wide spread and used a similar method of binning fall speed before data fitting (e.g., Barthazy and Schefold, 2006; Zawadzki et al., 2010). Shape and orientation affect the fall speed, since they are responsible for the drag force. Within most shape groups there is still a wide variety of different shapes. In addition, for any particle shape, the orientation may also contribute to the spread in data."

**16) Lines 187-190: Does this method of fitting use the scatter of fall speeds within each bin as an uncertainty for the bin fall speed (the median value)? It seems with such significant scatter in the individual data points, it would be important to de-emphasize bins with a lot of scatter when fitting. Also, if the bin widths become large (e. g., due to the requirement that all bins contain equal numbers of particles), it may be necessary to also treat scatter in the particle sizes within each bin as an uncertainty on the median particle size for each bin. In cases with uncertainties in both the x- and y-values to be fitted, a method such as orthogonal distance regression is needed.**

We have not used the scatter within each bin as a measure of uncertainty in the fitting method. Instead, we have selected our simpler fitting method.

When we started fitting after binning the data, we first tested equally-spaced Dmax or A bins, however, frequently observed issues with the first and/or last bin when only one or two particles were included in the bin. Changing bin sizes would then affect the fitted relationships. We could resolve these issues by requiring equal number of particles in each bin. The number of bins, ten, was a compromise we have selected between many bins but too few particles per bin and many particles in each bin but too few bins for a good fit.

**17) Lines 191-193: I'm not sure I agree completely with this assertion. Imagine an extreme case of fitting using M_b with only two bins. I wouldn't expect the results to match those from a M_a fit. Is there a reason that 10 bins were selected?**

As explained in 16), the number is a compromise to have enough bins to build a regression, but few enough so that there are enough particles in each bin to get a representative median to decrease the spread in data within each bin. With this compromise, we expect Mb and Ma to agree reasonably well, and in the following, we discuss for which shape groups this is the case and for which not. We will change the text to make this clearer.

MS changed as indicated and the number of bins, ten, has now been motivated in the MS in Line 187.

L. 191-193:
"Since the binned data is based on the individual data, the fits obtained from the binned data (Mb) should match the fits based on the individual data (Ma). When they do match, and, in particular, when R2 is high, the fits are deemed representative for the given shape group."

CHANGED TO:

"Since the binned data is based on the individual data, the fits obtained from the binned data (Mb) should be consistent with the fits based on the individual data (Ma). If so, and, in particular, when R2 for Mb is high, the fits are deemed representative for the given shape group."

L. 186-187: "…avoid the problem of individual bins having a disproportional effect on the fit. The binned data consists of the median values…"

CHANGED (by adding a sentence) TO:

"…avoid the problem of individual bins having a disproportional effect on the fit. The number of bins, ten, is a compromise; few enough bins to contain enough particles per bin and many enough bins to allow for a good fit to the measurements. The binned data consists of the median values…"

**18) Line 201-208: This is saying that (Rˆ2)_a is for v vs D_max, and (Rˆ2)_b is for v vs A. Is this what is intended? That doesn't seem consistent with what is shown in Tables 3 and 4. Instead, Tables 3 and 4 show that (Rˆ2)_a corresponds with M_a, and (Rˆ2)_b corresponds to M_b. My understanding, then, is that you want (Rˆ2)_b to be better than (Rˆ2)_a in order to believe that M_b provides a valid fit. Or do you want both correlations to be high? I think it is necessary to**

**review the discussion in these lines to make sure it says what you intended. See my opening comments.**

Yes, that was what we intended. And you are right it is inconsistent with Tab 3 and 4. See our response to 4) in general comments. In addition to the changes mentioned there, we will rephrase a few sentences here to make our intensions clearer, i.e. that we are looking at $R^2$ for Method b to check for which of the shape groups the fitted relationships for v vs Dmax and v vs A are reliable.

MS changed as indicated.

L. 199-201: "The method Ma fits agree reasonably well with Mb fits for all shape groups if considering confidence regions (see Figures A1 and A2 in Appendix A). Therefore, considering the correlation coefficients R2 from the binned data (Mb) is necessary in order to judge if the relationships are reliable or not."

CHANGED TO:

"The method Ma fits agree reasonably well with Mb fits for all shape groups if considering confidence regions (see Figures A1 and A2 in Appendix A). To judge if the relationships are reliable or not, the correlation coefficients RD2 (v vs Dmax) and RA2 (v vs A) for Mb will be considered too."

**19) Line 211-212: Given the data in Tables 3 and 4, it's not clear to me how binning improved both $(R^2)\_a$ and $(R^2)\_b$ for capped columns. $(R^2)\_a$ doesn't represent binned fits. See my opening comments.**

The confusion should be resolved by our response to 4) in general comments.
$R^2$ for v vs Dmax improved from 0.12 (Ma) to 0.42 (Mb). For v vs A the improvement is from 0.10 to 0.49.

**20) Lines 213-214: Similarly here, binning doesn't impact $(R^2)\_a$, so I'm not sure what is meant by "improved their correlation coefficients $(R^2)\_a...$". See my opening comments.**

See again our response to 4) in general comments.

**21) Line 223: Isn't cross-sectional area an aspect of particle size? Do you mean that fall speed depends more strongly on cross-sectional area than on maximum dimension? Figure 7 caption: Again, this usage of $(R^2)\_a$ and $(R^2)\_b$ seems inconsistent with how they are represented in Tables 3 and 4.**

We use particle size and maximum dimension interchangeably. For clarity, we will change "particle size" here to "maximum dimension" or change the text to make the interchangeability clearer. Regarding Fig. 7 caption, please see the answer to 4) in general comments.

MS changed as indicated. We have stated that particle size and maximum dimension are used synonymously at the beginning of Sect 2.2.

L. 99-100: "…maximum dimension, Dmax, defined as the smallest diameter that completely encircles the particle boundary in the top-view image, is used to describe the particle size."

CHANGED (by adding a sentence) TO:

"…maximum dimension, Dmax, defined as the smallest diameter that completely encircles the particle boundary in the top-view image, is used to describe the particle size. Thus, in the following, particle size and maximum dimension are used synonymously."

Where we had used "size" instead of "particle size",
we ADDED the word "particle" (e.g., see point 24) below).

**22) Lines 242: Does the "(15) Spherical" shape group then represent solid ice spheres?**

Shape group *(15) Spherical* includes frozen small raindrops and liquid raindrops. These particles have spherical shape and morphology but are solid or liquid particles as shown in the figure below (see more details in Vázquez-Martín et al., 2020 "Shape Dependence of Falling Snow Crystals' Microphysical Properties Using an Updated Shape Classification"). H1c are ice particles and H4a are liquid droplets. If the particle is a perfect sphere, it is assumed to be liquid and thereby considered a raindrop, whereas if the particle has a popping bubble on the edge, it is considered as an ice particle.

[Figure]

**23) Lines 272-274: For me, the most interesting aspect of this is that the vertically-oriented particles do not fall \*much\* faster than the horizontally-oriented particles. Per, for example, Mitchell and Heymsfield (2005), fallspeed should depend on the ratio of mass to horizontally-projected area. It would be interesting to see the comparisons shown in Figure 11, but resolved by particle shape. Do vertically-oriented plates fall faster than horizontally-oriented plates of the same D_max? I'm not suggesting that this be included in this paper, of course, but perhaps for future work.**

Thanks for your comment and suggestion, very interesting.
We had investigated what you mention for different shapes individually, i.e., for shape groups *(1) Needles, (5) Plates* and *(6) Stellar*. However, probably due to the small sample size, the results were not conclusive showing larger uncertainties than differences between the two falling orientations.
We will certainly look at these questions again in future studies when more data will be available.

**24) Line 285: I'm not sure what is meant by "overlapping somewhat in D_max and A".**

We will rephrase to make this clearer:

As can be seen in Fig. 12, these relationships are spread out in a way, so that for a given…

MS changed as indicated.

L. 285-286: "These relationships are overlapping somewhat in Dmax and A, respectively. Figure 12 illustrates that for a given size or cross-sectional area…"

CHANGED TO:

"As can be seen in Fig. 12, these relationships are spread out in a way, so that for a given particle size or cross-sectional area…"

**25) Lines 293-319: This analysis is highly interesting, but it took several read-throughs to follow the development. Since mass, area and area ratio can all be expressed succinctly using power laws as functions of D_max, I wonder if it might be clearer to develop, for example, the resulting power law relationships for mass and area as functions of area ratio then show, by substituting in the appropriate exponents for elongated particles, that mass increases more rapidly than does A as A_r increases. Is there a reason this approach wasn't used?**

The purpose of this section is to investigate the Ar dependence observed in Fig 12. For that, we are considering the special case of a subset of the data with constant Dmax. Because of that we cannot use the normal power laws as function of Dmax to do this investigation. If one would use them, then the particle mass were constant. We will modify the text slightly to improve clarity. See also our response to your comment 26), where we correct an error.

MS changed as indicated.

L. 300: "While A ∝ Ar is valid in general for all shapes…"

CHANGED TO:

"While, in case of constant $D_{max}$, A ∝ Ar is valid in general for all shapes…"

AND change described in point 26) below.

**26) Lines 308-311: If I look at prior studies like Mitchell (1996), it seems more common for other shapes to have mass power laws where the exponents are in the range of 1.7 to 2.4, rather than 3. Does this alter the interpretation presented here?**

We examine here the special case of constant Ar. For this A is proportional to $D^2$ as stated. However, it was wrong to state that m would then be proportional to $D^3$. There could of course be shapes such as simple plates where Ar is constant and where thickness (not seen on image) does not increase proportionally to D. In an extreme case, m would then be proportional to $D^2$. So, in general, in a dataset with constant Ar, m would be proportional to anything between $D^2$ to $D^3$. We will change the text to correct this. It does not alter the interpretation presented.

MS changed as indicated.

L. 308-310: "In this case, $A \propto D_{max}^2$, and $m \propto D_{max}^3$ in general. Consequently, as m increases much more rapidly with increasing $D_{max}$ than A, the fall speed also increases rapidly with increasing $D_{max}$, which is consistent…"

CHANGED TO:

"In this case, $A \propto D_{max}^2$ in general and $m \propto D_{max}^\beta$ where $\beta$ is between 2 and 3. Consequently, as m increases more rapidly with increasing $D_{max}$ than A (for all cases but the extreme $m \propto D_{max}^2$), the fall speed also increases rapidly with increasing $D_{max}$, which is consistent…"

**27) Figure 13 caption: Please also describe the purpose of the numbers labeling each line.**

The purpose of labelling each line was to easily connect the fit lines to the power laws displayed in Table 6. We will mention this in both captions of Table 6 and Fig. 13.

==MS changed as indicated in Table 6. In Fig. 13 this was already mentioned.==

In Table 6, we ADDED the following sentence to the end of the caption:
"To easily connect the fit lines to the power laws, the same relationship numbers have been used in Table 6 and Fig. 13."

**28) Table 6: For comparison purposes, I suggest that you consider converting all the relationships reported here to consistent units. I know it will affect only the leading coefficients, but it would make it easier to quantify the differences between the relationships.**

The reason why we have shown the relationships in different units was to be consistent with the power laws used from the literature. However, we agree that for comparison it would be more convenient if all used the same reference units. Therefore, we will be adding such information in the text, for instance, by adding "The power laws from the literature have been converted to the same units." Consequently, we will change the relationships shown in Table 6.

==MS changed as indicated including the mentioning of unit conversion.==

In Table 6, we have added on sentence to the caption after "The relationships found in this work are also shown as *[VM]*."

Which then CHANGED TO:

"The relationships found in this work are also shown as *[VM]*. The power laws from the literature have been converted to use the same units, i.e., mm and m $s^{-1}$, as in *[VM]*."

In the table itself, we changed the affected leading coefficients, the indicated units, and the "Reported range of D", e.g., for relationship 2.

FROM: "100–3000 µm        2. $v/(cm\ s^{-1}) = 184 \cdot (D/cm)^{0.56}$"

CHANGED TO:

"0.10–3.0 mm          2. $v/(m\ s^{-1}) = 0.51 \cdot (D/mm)^{0.56}$"

- ■ **We made some text edits:**

MS changed as indicated.

- • Line 102 has been corrected as it follows:

    "cross-sectional are" has been replaced by "cross-sectional area"

- • **Replace versus 'vs.' by 'vs'.**
We sincerely appreciate the referee #2 for your constructive feedback and time spent to read and evaluate this work. Please see below our response to your comments.

Changes done in the MS as a consequence of our responses are indicated; all line numbers refer to the MS as originally submitted.

- **General comments:**

**1) In the abstract, the points in the first "introduction" paragraph could be organized better and stated more succinctly, perhaps in about three sentences.**

Thank you for pointing this out. We will modify the first "Introduction" paragraph in the Abstract in order to make it more concise and also better organized (see also the response to specific comment 1 of referee 1).

MS changed as indicated.

The first "Introduction" paragraph has been shorten. Lines 3-4 have been removed. Then, lines 2-4: "Improved snowfall predictions require accurate knowledge of the properties of ice crystals and snow particles, such as their size, cross-sectional area, shape, and fall speed. In particular, the shape is an important parameter as it strongly influences the scattering properties of these ice particles, and thus their response to remote sensing techniques such as radar measurements."

CHANGED TO:

"Improved snowfall predictions require accurate knowledge of the properties of ice crystals and snow particles, such as their size, cross-sectional area, shape, and fall speed."

**2) Lines 43 – 49: The recent work of Dunnavan (2020, JAS) should also be mentioned here as he shows how orientation and other factors account for much of the dispersion in ice fall speed for a given size.**

Thanks for your suggestion. We will include that reference in the previous paragraph talking about the influence of the shape on the spread in fall speed, which in their study was more important than orientation.

MS changed as indicated.

Lines 38-40 (Sect 1) have been changed as follows:

"The physical properties of snow particles, including shape, govern their fall speed. For a given volume and density, non-spherical particles fall slower than spheres (Haider, 1989)."

CHANGED TO:

"The physical properties of snow particles, including shape, govern their fall speed. For a given volume and density, non-spherical particles fall slower than spheres (Haider, 1989). At the same size, shape variations account for spread in fall speed, which causes variations in other properties such as the vertical mass flux of water (Dunnavan, 2021)."

**3) Lines 50 – 52: Another reference that supports this is Mitchell et al. (2008, GRL), which shows how the cirrus ice fall speed affects cirrus cloud coverage and ice water path, and the resulting impact on radiation in the Community Atmosphere Model version 3 (CAM3).**

Thank you for your suggestion. We will add Mitchell et al. 2008.

MS changed as indicated.

Lines 50-52 (Sect 1) have been modified as follows:

"The fall speed of snow crystals plays a significant role in modelling microphysical precipitation processes (Schefold et al., 2002) and for climate since it determines the lifetime of cirrus clouds and influences the vertical transport of water vapour in the upper troposphere, modulating the top of atmosphere radiation budget (Westbrook and Sephton, 2017)."

CHANGED TO:

"The fall speed of snow crystals plays a significant role in modelling microphysical precipitation processes (Schefold et al.,2002) and for climate since it determines the lifetime of cirrus clouds, and thus the cloud coverage and ice water path (Mitchell et al., 2008), and the top of atmosphere radiation budget (Westbrook and Sephton, 2017)."

**4) Figure 3 and elsewhere in paper where shape category 13 is mentioned: Shape category 13 is ambiguous since it appears this can refer to any kind of ice particle and it also includes melting/sublimating ice particles of any shape apparently. Is this merely a "dust bin" category that includes all the other shapes that defy classification?**

Our "dust bin" is shape group *(14).* Group *(13)* is, as it says, melting and sublimating (mostly small) particles. However, we think you are right in that shape group *(14)* should have better been divided into "Irregulars" (i.e. non aggregates) and "Aggregates", that's something we did not consider when first doing the shape classification.

[Figure]

| I1 | H1a | H1b | H2a | H3a | 1.0 |

**5) Figure 3 and elsewhere in paper where shape category 14 is mentioned: Irregulars as defined in Lawson et al. articles are high density, blocky-type particles that are subdivided into small and large irregulars, but other authors may define them differently.**
**How are they defined here? If irregulars here are similar to irregulars in Lawson et al. papers, then grouping them with aggregates may be a mistake, since unrimed aggregates are NOT high density; rather they are notoriously low density. Please show a photo of irregulars so readers will know what they look like.**

Same as in 4), we think you are right in that shape group *(14)* should have better been divided into "Irregulars" (i.e. non aggregates) and "Aggregates", that's something we did not consider when first doing the shape classification. We have added pictures here and referred readers in the MS to our previous work on shape classification, which contains many example images (Vázquez-Martín et al, 2020).
In our dataset we do not have many aggregates yet. However, in future we will extend our dataset and then likely revise the shape grouping including a separate group for aggregates.

[Figure]

| A1a | A2a | A3a | A4a | A5a | A6a |
| A7a | A8a | A9a | A9b | A10a | A11a | A12a |
| A13a | A14a | I2 | I4 | G6 | 1.0 |

**6) Lines 175 – 181: Regarding the large spread in fall speeds, the recent paper by Dunnavan (2020, J. Atmos. Sci.) shows snowflake (i.e., crystal aggregate) fall speeds are most sensitive to particle shape differences (largely accounted for in this study) with secondary sensitivity to aggregate orientation. Please consider whether this Dunnavan study is relevant in explaining the large spread in fall speeds encountered here.**

We will mention the spread in fall speed caused by shape variations (Dunnavan 2020) in the Introduction in response to your general comment 2). Responding to specific comment 15) of Referee 1, we will also extend the discussion on the reasons for the fall-speed spread in the next section on the binning method.

MS changed as indicated.

Lines 38-40 (Sect 1) have been modified as follows:

"The physical properties of snow particles, including shape, govern their fall speed. For a given volume and density, non-spherical particles fall slower than spheres (Haider, 1989). Therefore, also the particle shape is an important parameter ..."

CHANGED TO:

"The physical properties of snow particles, including shape, govern their fall speed. For a given volume and density, non-spherical particles fall slower than spheres (Haider, 1989). At the same size, shape variations account for spread in fall speed, which causes variations in other properties such as the vertical mass flux of water (Dunnavan, 2021). Therefore, also the particle shape is an important parameter …"

Lines 189-190 (Sect 3.2.2) have been also modified as follows and considering the specific comment #15) from Referee 1:

"Previous studies have also used a similar method of binning fall speed before data fitting (e.g., Barthazy and Schefold, 2006; Zawadzki et al., 2010)."

CHANGED TO:

". The apparent randomness in fall speed, manifested as the wide spread in data, may have several reasons. While instrumental uncertainties and errors introduced by the manual analysis (see Kuhn and Vázquez-Martín, 2020) contribute to the variability, much of the observed randomness is likely inherent to the data. For example, Dunnavan (2021) showed that aggregate snowflakes' fall speed is very sensitive to shape. Other studies have also reported a wide spread and used a similar method of binning fall speed before data fitting (e.g., Barthazy and Schefold, 2006; Zawadzki et al., 2010). Shape and orientation affect the fall speed, since they are responsible for the drag force. Within most shape groups there is still a wide variety of different shapes. In addition, for any particle shape, the orientation may also contribute to the spread in data."

**7) Figure 6: The legend contains a pink bar denoting the 68% confidence region, but a pink region is not evident in the plots. Perhaps it is sufficient to only show the 68% prediction bands?**

This is indeed difficult to see. The confidence regions for Ma and Mb are shown as half-transparent pink and light blue, respectively. Here, they are almost entirely overlapping, making it difficult to make out where the pink region is. We will try to improve this, perhaps changing the v range to 0-1.0 m/s and removing the Mb prediction band will help. Otherwise, we will also consider your suggestion.

MS changed as indicated.

Figure 6 has been modified:

1) y-axis range has been changed. Range '0-1.8 m/s' replaced by '0-1 m/s'.
2) The line representing the 68% confidence region of Method Mb has been removed in both panels (v vs Dmax and v vs A). Consequently, the legend and the caption of the figure have been adapted.
3) 'vs.' has been also replaced by 'vs' as we pointed out in our responses to Referee 1. Note that the same correction has been done in the whole manuscript.

Then, caption in Figure 6 has been modified as follows:

"Figure 6. Fall speed versus particle size (v vs. Dmax ) and fall speed versus cross-sectional area (v vs. A) relationships for shape group (5) Plates. Individual data (brown symbols) and binned data (blue symbols with error bars) are displayed. Median values in the respective bins represent the binned data. The total length of the error bars represents the spread in fall speed data, which is given by the difference between the 16 th and 84 th percentiles. Fits that apply to individual data (Ma) and to binned data (Mb) are shown for comparison. The 68% prediction band and the 68% confidence region for both fits (Ma, Mb) are also shown. Left: v vs. D max relationship given by Eq. (4). Right: v vs. A relationship given by Eq. (5). The same data are shown in Table 3 for v vs. D max and in Table 4 for v vs. A."

CHANGED TO:

"Figure 6. Fall speed versus particle size (v vs Dmax) and fall speed versus cross-sectional area (v vs A) relationships for shape group (5) Plates. Individual data (brown symbols) and binned data (blue symbols with error bars) are displayed. Median values in the respective bins represent the binned data. The total length of the error bars represents the spread in fall speed data, which is given by the difference between the 16 th and 84 th percentiles. Fits that apply to individual data (Ma) and to binned data (Mb) are shown for comparison. The 68% prediction band for both fits (Ma, Mb) are shown. The 68% confidence region is shown for Mb. Left: v vs D max relationship given by Eq. (4). Right: v vs A relationship given by Eq. (5). The same data are shown in Table 3 for v vs D max and in Table 4 for v vs A."

**8) Section 3.3.1 on Orientation: A study by Kajikawa is the only one I know of containing observational results on natural ice crystal orientation during free-fall; please include this reference (Kajikawa, 1992, J. Meteorol. Soc. Japan), comparing orientation results from this study with the Kajikawa results.**

Thank you, this is an interesting study. Kajikawa 1992 looked at falling motion with stable and unstable patterns, rather than at orientation directly as we did, and found only small variations (few %) in the vertical speed with respect to the mean for individual trajectories. The smaller horizontal speeds showed more variations that Kajikawa linked to growth by aggregation.

**9) Section 3.4 (Comparison with previous fall speed relationships): Ice fall speeds depend on temperature T and pressure P; what were the values assumed here? Was this assumption applied universally to all of the studies in this intercomparison? If not, please ensure that all fall speeds in this intercomparison use the same T and P.**

Thank you for raising this issue. We are aware of this fact and that different studies are based on measurements at different atmospheric conditions and may be adjusted to some normal conditions or not. Our measurement site is at an altitude of around 400 m, which corresponds to about a pressure difference of 50 hPa with respect to sea level. A pressure difference of 100 hPa (about 1000 m altitude difference) would result in about 5% difference in fall speed (see for example Khvorostyanov and Curry 2005, JAS 62, 4343-4357). A temperature difference of 10 K would cause a 2% speed difference. As these differences are relatively small compared to the typical speed differences between different studies we decided, for simplicity, to neglect these effects without any impact on the comparison. We will mention the altitude of our site and make a statement about the neglected P and T dependence of fall speed in the MS.

MS changed as indicated in Sect 2.1 (mentioned altitude), Sect 2.2 (mentioned that measurements not adjusted), and Sect 3.4 (mentioned that previous studies not adjusted to common conditions).

Lines 76-77 (Sect. 2.1) have been modified as follows:

"Our measurements are carried out in Kiruna, Sweden (67.8º N, 20.4º E) using D-ICI, the ground-based in-situ instrument described in Kuhn and Vázquez-Martín (2020). D-ICI captures and records ..."

CHANGED TO:

"Our measurements are carried out in Kiruna, Sweden (67.8º N, 20.4º E, at approximately 400 m above sea level), at a site described in Vázquez-Martín et al. (2020), using D-ICI, the ground-based in-situ instrument described in Kuhn and Vázquez-Martín (2020). D-ICI captures and records ..."

Lines 113-115 (Sect 2.2) have been modified as follows:

"The falling distance is then the average of the euclidean distances between P1 and P2, and between P3 and P4, and the fall speed is this falling distance divided by the time between exposures. By selecting at least two points on each particle ..."

CHANGED TO:

"The falling distance is then the average of the euclidean distances between P1 and P2, and between P3 and P4, and the fall speed is this falling distance divided by the time between exposures. These fall speeds are reported as they are measured at our local conditions and are not corrected to, for example, sea level pressure, which would only change values by less than 3%. <NEW PARAGRAPH>

By selecting at least two points on each particle ...”

Lines 328-329 (Sect 3.4) have been modified as follows:

“D corresponds to the maximum length of any horizontal row in the side-view shadowgraphs. *[Lo]* studied fall speeds ...”

CHANGED TO:

“D corresponds to the maximum length of any horizontal row in the side-view shadowgraphs. Furthermore, we have not adjusted the different studies to common temperature and pressure conditions but compared them as they are reported. While some did adjust measurements to some standard conditions, others did not. For example, *[H]* adjusted measurements from about 1000 m altitude to a pressure level of 1000 hPa, whereas *[Lo]* used measurements from, on average, the same altitude but did not adjust them to a common or standard pressure level, which results in a difference of about 5%. <NEW PARAGRAPH>
*[Lo]* studied fall speeds ...”

**10) Figure 13, middle panel (stellar): Please mention that Curve 11 is based on the flow regime for larger ice particles, and should be closer to [VM] for the smaller D if the corresponding flow regime constants in [M] were used.**

Thank you for pointing this out. We will mention that in the MS.

MS changed as indicated.

Lines 349-350 (Sect 3.4) have been modified as follows:

“As for plates, also for stellar particles the previous relationships by *[H]* and *[M]* are closest to *[VM]*. Again, *[Le]* and *[B]* reported relationships ...”

CHANGED TO:

“As for plates, also for stellar particles the previous relationships by *[H]* and *[M]* are closest to *[VM]*. Note, that *[M]* is based on the flow regime for particles larger than about 1 mm (Eq. 20 in *[M]*). Using the flow regime for smaller particles, *[M]* would come somewhat closer to *[VM]* below about 0.6 mm. Again, *[Le]* and *[B]* reported relationships ...”

**11) Lines 343-4: Sentence structure appears awkward; do you mean to say "Our shape group (5) contained only plates that were unrimed"?**

The sentence will be modified accordingly and convey a little more detail.

MS changed as indicated.

Lines 343-344 (Sect 3.4) have been modified as follows:

"Our data included in shape group *(5) Plates* are unrimed. However, even the unrimed plates ..."

CHANGED TO:

"Our data included in shape group *(5) Plates* are mainly composed of unrimed particles (for a detailed description, see Vázquez-Martín et al., 2020). However, even the unrimed plates …"